# Past Ice-Sheet Behaviour:
# Retreat Scenarios and Changing Controls in the Ross Sea, Antarctica

Anna Ruth W. Halberstadt[1], Lauren M. Simkins[1], Sarah L. Greenwood[2], John B. Anderson[1]

[1]Department of Earth Science, Rice University, Houston, Texas 77005, USA

[2]Department of Geological Sciences, Stockholm University, 10691 Stockholm, Sweden

*Correspondence to*: A. R. Halberstadt (ar.halberstadt@rice.edu)

**Abstract.** Studying the history of ice-sheet behaviour in the Ross Sea, Antarctica's largest drainage basin, can improve our understanding of patterns and controls on marine-based ice-sheet dynamics and provide constraints for numerical ice-sheet models. Newly collected high-resolution multibeam bathymetry data, combined with two decades of legacy multibeam and seismic data, are used to map glacial landforms and reconstruct paleo ice-sheet drainage.

During the Last Glacial Maximum, grounded ice reached the continental shelf edge in the eastern but not western Ross Sea. Recessional geomorphic features in the western Ross Sea indicate virtually continuous back-stepping of the ice-sheet grounding line. In the eastern Ross Sea, well-preserved linear features and a lack of small-scale recessional landforms signify rapid lift-off of grounded ice from the bed. Physiography exerted a first-order control on regional ice behaviour, while seafloor geology played an important subsidiary role.

Previously published deglacial scenarios for Ross Sea are based on low-spatial-resolution marine data or terrestrial observations; however, this study uses high-resolution basin-wide geomorphology to constrain grounding-line retreat on the continental shelf. Our analysis of retreat patterns suggests that: (1) retreat from the western Ross Sea was complex due to strong physiographic controls on ice-sheet drainage; (2) retreat was asynchronous across Ross Sea and between troughs; (3) the eastern Ross Sea largely deglaciated prior to the western Ross Sea following the formation of a large grounding-line embayment over Whales Deep; and (4) our glacial geomorphic reconstruction converges with recent numerical models that call for significant and complex East Antarctic Ice Sheet and West Antarctic Ice Sheet contributions to the ice flow in the Ross Sea.

## 1 Introduction

The Ross Embayment drains ~25% of the Antarctic Ice Sheet into the Ross Sea and is thus the largest ice drainage basin in Antarctica, fed by multiple ice streams sourced from the East Antarctic (EAIS) and West Antarctic (WAIS) ice sheets (Fig. 1). The nature of ice-sheet paleodrainage and retreat in the Ross Sea has significant implications for understanding the

dynamics of the WAIS and EAIS, and their respective sensitivities to factors that govern ice behaviour. These insights may also aid understanding of ice dynamics in the other large embayments around Antarctica, such as the Weddell Sea and Amundsen Sea Embayments, where large uncertainty in paleo-ice extent and grounding-line retreat remains (Bentley et al., 2014). Recent ice-sheet models indicate complex ice behaviour in the Ross Sea, particularly during deglaciations (e.g.

Pollard and DeConto, 2009; Golledge et al., 2014; DeConto and Pollard, 2016). Geologic reconstructions of ice dynamics from the Ross Sea continental shelf can provide critical tests for these models.

Multibeam bathymetry provides a direct record of bed conditions beneath the former ice sheet, revealing landforms associated with past ice flow. These landforms document the flow behaviour of formerly grounded ice. Here we compile legacy multibeam bathymetry data from 41 cruises over the last 20 years (Supplementary Table 1), combined with recently

acquired high-resolution multibeam data, to characterize glacial geomorphic features across the Ross Sea. This unique, integrated dataset provides an opportunity to view the paleo ice-sheet bed at a much higher resolution than is possible beneath the modern ice shelf and ice sheet. We can improve our understanding of factors that control regional ice-sheet dynamics and test existing ice-sheet retreat models by using this dataset to identify glacial geomorphic features that characterize past flow and retreat dynamics. These geomorphic features are used to reconstruct ice-sheet paleodrainage

across the Ross Sea during and subsequent to the LGM.

## 2 Study Area

The Ross Sea contains seven bathymetric troughs (Fig. 1), which are remnants of the extensional tectonic history of the region (Lawver et al., 1991). Ice streams preferentially occupied these troughs and eroded along pre-existing tectonic lineaments, scouring the seafloor over multiple glacial cycles (Cooper et al., 1991; Anderson, 1999). The eastern Ross Sea

(ERS) and the western Ross Sea (WRS) have distinctly different characteristics in terms of seafloor geology and physiography. The WRS is geologically complex with older and more consolidated strata locally occurring at or near the seafloor (Cooper et al., 1991; Anderson and Bartek, 1992). The WRS contains high-relief banks and deep troughs, and thus serves as an analogue to the modern Siple Coast grounding line where banks currently serve as ice rises and provide a buttressing effect to the grounding line (e.g. Matsuoka et al., 2015). The ERS is dominated by a single, large rift basin,

bounded by Ross Bank and Marie Byrd Land, with near-surface stratigraphy comprised of unconsolidated Plio-Pleistocene sediments that thicken in a seaward direction (Alonso et al., 1992; De Santis et al., 1997). The ERS has more subdued physiography consisting of broad troughs separated by low-relief ridges (Fig. 1).

Results from marine geological research, including integrated seismic stratigraphy, geomorphology, and sediment core analyses, indicate that both the EAIS and WAIS advanced across the continental shelf during the Last Glacial Maximum

(LGM; Licht et al., 1999; Shipp et al., 1999; Mosola and Anderson, 2006; Anderson et al., 2014). The relative contributions

of the EAIS and WAIS to LGM ice flow and subsequent paleodrainage and retreat behaviour in the Ross Sea remain controversial. Results from several land-based studies have led to the conclusion that the WAIS dominated ice flow during the LGM (e.g. Denton and Marchant, 2000; Hall and Denton, 2000; Hall et al., 2015). However, offshore till provenance analyses indicate that the EAIS and WAIS had roughly equal contributions to ice draining into the Ross Sea (Anderson et al.,

1984; Licht et al., 2005; Farmer et al., 2006). Significant drainage of the EAIS into the western Ross Sea is also supported by interpretations from seafloor glacial geomorphology (Shipp et al., 1999; Mosola and Anderson, 2006; Greenwood et al., 2012; Anderson et al., 2014), exposure age dating in the Transantarctic Mountains (e.g. Jones et al., 2015), and numerical ice-sheet models (e.g. Golledge et al., 2013, 2014; McKay et al., 2016; DeConto and Pollard, 2016).

Based on the WRS continental shelf record, the Drygalski Trough grounding line is thought to have stepped back from its

LGM position south of Coulman Island by ~13.0 cal ka and reached Drygalski Ice Tongue by ~11.0 cal ka (Licht et al., 1996; Cunningham et al., 1999; McKay et al., 2008; Anderson et al., 2014). In Terra Nova Bay, however, just north of Drygalski Ice Tongue, radiocarbon dates from raised beaches place the establishment of ice-free conditions at ~8.2 ka. The grounding line retreated into McMurdo Sound at ~7.7-7.8 ka, based on a radiocarbon-dated marine shell (*Adamussium colbecki*; Licht et al., 1996), ice-dammed lakes (Hall and Denton, 2000), and relative sea-level records (Hall et al., 2004;

2013). Ages from sediment cores collected by McKay et al. (2008) place grounding-line retreat in McMurdo Sound at ~10.0 ka. More recent results from McKay et al. (2016) indicate that the grounding line may have reached the vicinity of Ross Island prior to ~8.6 cal ka, although a relative sea-level record from raised beaches on the southern Scott Coast suggest final unloading of grounded ice at ~6.6 ka (Hall et al., 2004). In general, land-based ages of deglaciation lag behind the marine record (Anderson et al., 2014), suggesting that either marine grounding-line retreat may have preceded continental ice

thinning, and/or that grounding-line retreat proceeded westward from the WRS towards the coast.

Dynamic ERS ice-stream behaviour has been hypothesized, including pre-LGM retreat and subsequent re-advance (Bart and Owolana, 2012). ERS marine radiocarbon ages suggest very early retreat from the continental shelf break during or before the LGM (Licht and Andrews, 2002; Mosola and Anderson, 2006; Bart and Cone, 2012; Anderson et al., 2014), although methods for obtaining these dates remain highly problematic due to possible reworking of old carbon (Licht and Andrews,

2002) and uncertainties of appropriate marine reservoir corrections (Hall et al., 2010). Conversely, terrestrial studies of ice-sheet thinning and measurements of post-glacial rebound in Marie Byrd Land indicate that ERS deglaciation occurred throughout the Holocene (Stone et al., 2003; Bevis et al., 2009). A comprehensive review of Ross Sea deglaciation is provided by Anderson et al. (2014), reviewing the extensive work that has been done in this region. Outstanding challenges in the Ross Sea include integrating and improving marine and terrestrial chronologies, as well as constraining the

contributions of the EAIS and WAIS to ice flow in the Ross Sea, their respective behaviour, and their sensitivity to various forcings. Here we use the Ross Sea-wide glacial geomorphological record to reconstruct the regional pattern of deglaciation and provide a spatial framework for interpreting point-sources of information such as cores and ages.

## 3 Methodology

This study synthesizes multibeam datasets from across the Ross Sea, combining legacy data (Supplementary Table 1) with newly collected, high-resolution data collected in key areas for characterizing the nature of ice-sheet retreat (Fig. 2a). The combined ship tracklines across Ross Sea cover over 250,000 km, providing unparalleled coverage of multiple paleo-ice

streams. New, high-resolution multibeam bathymetry data were acquired during an RV/IB *Nathaniel B. Palmer* NBP1502A cruise to the Ross Sea in the 2014-2015 Austral summer. These data were collected with a Kongsberg EM-122 system in dual swath mode with a 1°x1° array, 12 kHz frequency, and gridded at 20 m. Vertical resolution varies from about 0.2-0.07% of water depth (Jakobsson et al., 2011); therefore, at water depths of 500 m, geomorphic features with sub-meter amplitudes can be resolved. Horizontal resolution is similarly depth-dependent and, in water depths of 500 m, is about 9 m.

Ping editing was completed onboard using CARIS and imported into ArcGIS. In addition to multibeam data, newly acquired high-frequency seismic data (3.5 kHz sub-bottom data collected with a Knudsen CHIRP 3260 using a 0.25 ms pulse width) were interpreted along with legacy CHIRP data.

The seafloor geologic setting has been recorded in legacy seismic reflection data across the Ross Sea. We refer to seismic records as either 'high-frequency' denoting 3.5 kHz CHIRP data or 'low-frequency' referring to traditional seismic data (20-

600 Hz). Low-frequency seismic lines from cruise PD-90, originally published in Anderson and Bartek (1992), were combined with ANTOSTRAT Project seismic lines compiled by Brancolini et al. (1995). These previous investigators recognized seaward thickening units bounded by glacial unconformities, where each surface represents a glacial advance that eroded the previous substrate and deposited till and glacimarine sediments above the newly formed surface.

Glacial geomorphic features imprinting the Antarctic continental shelf are divided into three main categories, largely

following the classification scheme of Benn and Evans (2010). These are: (1) subglacial features, such as mega-scale glacial lineations (MSGLs), drumlinoid features, and subglacial channels; (2) ice-marginal features, such as grounding zone wedges (GZWs), marginal moraines, and linear iceberg furrows; and (3) proglacial features, such as gullies and arcuate iceberg furrows (Fig. 2). These features occur above the most recent shelf-wide glacial unconformity (with the exception of drumlinoids) and are covered by post-glacial sediments. They are, therefore, interpreted as features formed during the last

glacial cycle (e.g. Shipp et al., 2002; Mosola and Anderson, 2006).

### 3.1 Subglacial Features

Subglacial features form beneath permanently grounded ice that is thick enough to offset buoyant forces exerted by the ocean. MSGLs (Fig. 2b), the most common subglacial features on the Antarctic continental shelf, are streamlined features with high parallel conformity (Clark, 1993). While the actual formation process for MSGLs is still debated (e.g. Tulaczyk et

al., 2001; Shaw et al., 2008; Ó Cofaigh et al., 2008; Fowler, 2010), they are interpreted as having formed under streaming ice

due to their association with modern ice streams (King et al., 2009) and their occurrence within paleo-glacial troughs (Anderson, 1999; Livingstone et al., 2012). The streamlined nature of MSGLs makes them excellent indicators of ice-flow direction (Clark, 1993; Stokes and Clark, 1999; Shipp et al., 1999; Ó Cofaigh et al., 2002; Dowdeswell et al. 2004; Spagnolo et al., 2014). Previous studies have shown that MSGLs are associated with a massive seismic facies interpreted as the deformation till layer deposited above the latest glacial unconformity (Shipp et al., 1999; Ó Cofaigh et al., 2002, 2005; Heroy and Anderson, 2005). Most MSGLs in the Ross Sea have amplitudes of 1-9 m, and lengths of about 1-10 km. As these features are associated with deforming till, MSGL amplitudes should not be greater than the thickness of the deforming till layer.

Smaller scale streamlined features, with lengths of hundreds of metres to a few kilometres, comprise a number of landform classes such as drumlins, crag and tails, and megaflutes. We group these landforms here as a single class of drumlinoids. While their internal composition can be difficult to determine in the marine environment, and their formation mechanisms remain uncertain, this family of landforms is widely and most simply taken to record the former ice flow direction (Benn and Evans, 2010). In Antarctica, drumlinoids are most often observed at the transition between crystalline bedrock and sedimentary deposits (Wellner et al., 2001, 2006; Graham et al., 2009).

Subglacial meltwater channels have been reported from a number of locations on the Antarctic continental shelf, though almost exclusively incising crystalline bedrock on the inner shelf (e.g. Lowe and Anderson, 2003; Anderson and Fretwell, 2008; Smith et al., 2009; Nitsche et al., 2013; Witus et al., 2014). Channels in sedimentary substrates are rare, but have been previously observed on the Ross Sea continental shelf by Alonso et al. (1992), Wellner et al. (2006), and Greenwood et al. (2012), though their origin and link to subglacial meltwater is not evident.

**3.2 Ice-Marginal Features**

Ice-marginal features form at the grounding line, marking the transition from permanently grounded ice to ice that has decoupled from its bed to become a floating ice shelf or a calving ice cliff. They include GZWs, marginal moraines, and linear iceberg furrows.

GZWs are depositional features (Fig. 2f, 2g), characterized by relatively steep foreset slopes that result in asymmetrical morphologies, broadly indicating ice-flow direction during GZW deposition. GZWs are formed during periods of stability of the grounding line. They grow as sediment is delivered to the grounding line through subglacial bed deformation and basal debris melt-out (e.g. Alley et al., 1986, 1989; Anderson, 1999; Anandakrishnan et al., 2007; Alley et al., 2007; Dowdeswell et al., 2008, Dowdeswell and Fugelli, 2012; Batchelor and Dowdeswell, 2015). The growth of GZWs can stabilize an ice sheet against small-scale relative sea-level rise and ice-sheet thinning by reducing the minimum ice thicknesses necessary to counter buoyancy effects (Alley et al., 2007). Large GZWs can imply longer episodes of stability of the ice margin (Alley et

al., 2007, Dowdeswell and Fugelli, 2012). Here, GZWs are grouped into three categories: small-scale, intermediate-scale, and large-scale. Small-scale GZWs have heights less than 10 m, cannot be traced across an entire trough width, and generally are only observable in high-resolution multibeam and side-scan sonar data (e.g. Shipp et al., 1999; Jakobsson et al., 2012; Simkins et al., in press). Intermediate-scale GZWs range from 10-50 m heights and often display very sinuous fronts.

Large-scale GZWs (Fig. 2g) have heights exceeding 50 m and extend across the entire trough width. The internal structure of large GZWs is occasionally detectable in low-frequency seismic data and includes distinct foreset beds indicative of GZW progradation (e.g. Anderson, 1999; Heroy and Anderson, 2005; Dowdeswell and Fugelli, 2012), but more often internal reflectors are not resolved in seismic data (e.g. Mosola and Anderson, 2006; Batchelor and Dowdeswell, 2015).

Ice-marginal moraines (Fig. 2e) are often symmetric in cross section (Dowdeswell and Fugelli, 2012), but can also display a

slightly asymmetric shape (Winkelmann et al., 2010; Klages et al., 2013). They are generally believed to be formed by push-processes (e.g. Batchelor and Dowdeswell, 2015). Low-amplitude (<5 m) features with similar characteristics are sometimes interpreted as De Geer moraines, whose development is influenced by seasonal or cyclic processes (Hoppe, 1959; Lindén and Möller, 2005; Todd et al., 2007), or transverse ridges (Dowdeswell et al., 2008) which does not imply seasonal formation. Due to their limited amplitudes, these features are best resolved with high-resolution bathymetric mapping

techniques, such as the EM122 multibeam system and side-scan sonar (e.g. Shipp et al., 1999; Jakobsson et al., 2011; Simkins et al., in press).

Linear iceberg furrows exhibit high parallel conformity, often display a geomorphic expression similar to MSGLs, and are consistent with MSGL orientations. These linear furrows, however, are erosional features whereas MSGLs are interpreted as either depositional or deformational features. The margins of marine-based ice sheets with low-slope profiles are particularly

susceptible to tidal fluctuations, causing large areas of the ice sheet to intermittently contact the seafloor (Fricker and Padman, 2006; Brunt et al., 2010). We interpret linear furrows to form within this diffuse grounding zone, where ice is hovering at the buoyancy limit and cyclically contacting the seafloor. Alternatively (or additionally), linear furrows may be associated with ice-shelf breakup events when icebergs near the grounding line are held upright within an iceberg armada (MacAyeal et al., 2003; Jakobsson et al., 2011, Larter et al., 2012). Small repeating corrugation ridges have been observed

within fields of iceberg furrows or within individual iceberg furrows (Figs. 2h, 2k; Anderson, 1999; Jakobsson et al., 2011, Klages et al., 2015). Although the exact mechanism for their formation remains somewhat controversial, corrugation ridges are thought to form as icebergs move vertically with tides, causing iceberg keels to intermittently contact the bed (Jakobsson et al., 2011; Graham et al., 2013). Their association with vertical tidal movement is based on the occurrence of identical features in proglacial arcuate iceberg furrows (Anderson, 1999) and comparison of corrugation amplitude and spacing with

tidal modelling results (Jakobsson et al., 2011). Similarly, a deep keel capable of ploughing a linear furrow once ice has fractured could also have existed as an irregularity at the ice base prior to calving, forming linear furrows in a diffusive grounding zone. Both mechanisms for linear furrow formation signify that ice was still moving as a coherent body in contact

with the seafloor. We argue that the linear forward motion that ploughs these furrows is caused by upstream ice flow; therefore, their significance for ice-flow direction is the same as MSGLs. For this reason, linear furrows and MSGLs are grouped into the inclusive term of 'linear features.'

## 3.3 Proglacial Features

Shelf-edge gullies on high-latitude continental margins occur where streaming ice reached the continental shelf break (Fig. 2i). Although their origin remains uncertain, they have been attributed to many formative processes, including point sources of sediment-dense meltwater from the grounding line when it was situated at the shelf break (Anderson, 1999; Evans et al., 2005), or from small-scale slope failure due to accumulation of proglacial sediment (Gales et al., 2012). Both mechanisms imply proximity to the grounding line. Ross Sea shelf-edge gullies have not been extensively surveyed; however, a lack of
significant sediment infilling suggests that they were active during the LGM (Shipp et al., 1999).

Arcuate iceberg furrows (Fig. 2j) are common features near continental shelf margins and on bank tops, overprinting any potentially pre-existing landforms. These are clearly proglacial features formed by freely moving icebergs that drifted under the influence of ocean currents and winds. Corrugation ridges have been observed within arcuate iceberg furrows, which is the most compelling evidence that these ridges result from tidal motion (Anderson, 1999).

## 4 Results

The landform categories set out above were mapped from our composite multibeam dataset and their distributions are presented in Figure 3. We find significant differences in landform assemblage composition and distribution between the WRS and ERS.

### 4.1 Western Ross Sea

Drygalski Trough is the deepest region of the Ross Sea with water depths over 1000 m. Within this trough, the most seaward geomorphic expression of the ice-sheet grounding line is a large-scale GZW north of Coulman Island (D1, Fig. 3). This is consistent with previous interpretations of the maximum grounding line location (Licht et al., 1999; Shipp et al., 1999). A prominent set of MSGLs extends continuously from the Drygalski Ice Tongue to the approximate latitude of Coulman Island. A few small GZWs occur along the flanks of the trough; otherwise the MSGLs are not overprinted by recessional
features. North of Coulman Island, both linear and arcuate iceberg furrows overprint GZW D1. The outermost shelf is covered by extensive arcuate iceberg furrows, which could have overprinted any older features.

Multibeam data are scarce in southern Drygalski Trough, a key area for reconstructing the final phase of deglaciation in the WRS. Available data show a field of closely spaced, small-scale GZWs that back-step up the southern margin of Crary

Bank, and a set of discrete intermediate-scale GZWs and lineations offshore of Mackay Glacier that record westward grounding line retreat (Greenwood et al., 2012; Anderson et al., 2014).

JOIDES Trough is slightly fore-deepened on the outer shelf, relatively flat on the middle shelf, and slopes steeply into the deep inner shelf Central Basin (Figs. 1, 3). The outer portion of JOIDES Trough is mostly devoid of linear features, with the exception of one group of straight furrows. High-frequency CHIRP data show a 4-8 m thick layer of acoustically laminated and draped glacimarine sediments on the outer shelf (Fig. 3a). LGM-age carbonates occur on outer shelf banks on both sides of JOIDES Trough (Taviani et al., 1993; Fig. 3), precluding the presence of grounded ice at those locations. A large-scale, mid-shelf GZW (J1, Fig. 3) is seismically resolved (Shipp et al., 1999), although the GZW crest lacks clear expression in multibeam data. This mid-shelf GZW is separated from the next intermediate-scale GZW near the southern end of Crary Bank (J2, Fig. 3) by an extensive field of iceberg furrows followed by a continuous field of marginal moraines and small-scale GZWs. Southern JOIDES Trough is characterized by a series of meltwater channels associated with GZW erosional notches, observed at GZW J2 and ice-marginal features south of J2. The channels are incised into till deposited above the LGM unconformity and are occasionally overprinted by marginal moraines, therefore they are interpreted as subglacial channels that were active during the most recent glacial recession.

Pennell Bank and Ross Bank are linked across Pennell Trough by a bathymetric high (referred to here as Pennell Saddle), separating the outer shelf part of Pennell Trough from the deep Central Basin (Fig. 3). A large-scale GZW (P1) occurs at the northern margin of the Pennell Saddle and a thick (up to 14 m) package of layered glacimarine sediments extends northward from beneath the toe of the P1 GZW (Figs. 3b, c). Small-scale sinuous GZWs and relatively straight-crested moraines record the grounding line back-stepping from atop Pennell Saddle southward into Central Basin. These recessional features overprint a large subglacial meltwater channel within the saddle (Fig. 2d).

The Central Basin is a bathymetric low that reaches water depths of over 1000 m, situated south of all three WRS troughs. It contains multiple generations of poorly preserved linear features, suggesting phases of large-scale ice stream flow reorganization through the basin and McMurdo Sound (Greenwood et al., 2012). Numerous pockets of small, marginal moraines are found throughout the Central Basin and do not seem to be oriented parallel to depth contours.

## 4.2 Eastern Ross Sea

The ERS contains three major troughs (Glomar Challenger Basin, Whales Deep, and Little America Basin) separated by low-relief ridges that are thought to have separated three paleo-ice streams (Mosola and Anderson, 2006). Linear features dominate the ERS seafloor and extend to the continental shelf break (Fig. 3). They are associated with GZWs that are large enough to be identified in low-frequency seismic reflection data (Mosola and Anderson, 2006), (Fig. 2g; Fig. 3). Small- and intermediate-scale GZWs and moraines are confined to a few locations and no subglacial channels have been observed in the

ERS. Shelf-edge gullies occur at the continental shelf break, implying the delivery of sediment and meltwater to a shelf-break grounding-line position.

The only drumlinoids observed in the Ross Sea occur on the inner shelf of Glomar Challenger Basin (Fig. 2c), covering ~300 km$^2$, and are associated with a near-surface occurrence of crystalline bedrock (Anderson, 1999; Shipp et al., 1999). Because these features are moulded predominantly from bedrock, they likely formed over multiple glacial cycles. They do, however, exhibit highly uniform orientations (Fig. 2c) that are consistent with MSGL orientations seaward of the drumlinoids, indicating that the most recent phase of ice flow was likely responsible for the final drumlinoid shape. Extensive linear features occur throughout Glomar Challenger Basin (Fig. 2b, 3). They exhibit both trough-parallel and sub-parallel orientations, and are partitioned into discrete clusters based on orientation (see below). Legacy high-frequency CHIRP data in outer Glomar Challenger Basin show thin glacimarine sediments (Fig. 3d) and sediment cores sampled tills that typically occur within 1 to 2 meters of the seafloor. Two closely spaced large-scale GZWs exist at the continental shelf break, observed in low-frequency seismic lines (G1 and G2, Fig. 3). These GZWs are wide and long but relatively thin so that they are not clearly observable in multibeam bathymetry. Two large-scale composite GZWs on the mid-to-inner shelf (G3, G4) are observed in both low-frequency seismic and multibeam data (Bart and Cone, 2012).

Whales Deep also contains a large-scale GZW at the continental shelf break (W1, Fig. 3), observed only in seismic data, as well as a mid-shelf GZW observable in both low-frequency seismic and multibeam records (W2, Fig. 3). A well-developed field of linear features extends from beneath the mid-shelf (W2) GZW to the continental shelf break. Linear features are notably absent south of W2. Little America Basin, like Glomar Challenger Basin, exhibits extensive linear features that extend across the entire trough to the shelf break. Three large-scale GZWs (L1-3, Fig. 3) are identified from low-frequency seismic data (Mosola and Anderson, 2006), but are too relatively thin to be observed in the legacy multibeam bathymetry, which has limited coverage and quality in this area.

### 4.2.1 ERS Flowsets

Different flow directions in the ERS can clearly be identified by the presence of multiple generations of overprinting linear features. Discrete flow episodes, corresponding to the formation of distinct sets of linear features, are defined from the population of linear features in the ERS. Linear features were grouped based on their parallel concordance, close proximity, and similar morphometry (cf. Clark, 1999). Rose diagrams were constructed from each group of linear features to confirm that features within a flowset have similar orientations (Fig. 4). The orientation of linear features within a single flowset deviate by generally less than 10°, and thus each flowset is assumed to represent a single flow configuration whose component lineations were formed contemporaneously (cf. Clark, 1999). Assuming that all flowsets were shaped during and subsequent to the LGM, a relative chronology of their formation can be assessed based on their landward succession and

cross-cutting relationships with other flowsets. In order to characterize large-scale regional flow patterns, flowsets with discrete yet similar orientations were assumed to reflect a similar ice-flow configuration and grouped together for analysis.

Our new compilation of multibeam data reveals that major flow patterns in the ERS often deviate from the trough-parallel drainage that has been described previously (Licht et al., 2005; Mosola and Anderson, 2006; Anderson et al., 2014). Some flowsets in Glomar Challenger Basin exhibit evidence of trough-parallel flow (flowsets *a-c*, Fig. 4), but other flowsets indicate flow across an inter-ice-stream ridge towards Whales Deep (flowsets *d-h*, Fig. 4). Flowset *g* contains the only curvilinear flowlines observed. For this flowset, rose diagrams were used to exclude the possibility that the curvature indicates two discrete flow events with similar orientation. In Whales Deep, only one flowset is observed, consisting of trough-parallel features on the outermost shelf (flowset *I*, Fig. 4). Flow indicators in Little America Basin resemble the configuration in Glomar Challenger Basin: some linear features in Little America Basin record trough-parallel flow (flowsets *j*, *k*, Fig. 4), while others are oriented oblique to the trough axis, pointing towards Whales Deep (flowsets *l*, *m*, Fig. 4). A third group of linear features indicates flow out of Little America Basin into a neighboring outlet draining Marie Byrd Land to the east (flowset *n*, Fig. 4). Flowsets on the innermost shelf in all three ERS troughs are interpreted to indicate late-stage deglacial flow configurations.

## 5 Discussion

### 5.1 Last Glacial Maximum ice extent and flow

We interpret the LGM grounding line in outer Drygalski Trough to have been situated just north of Coulman Island, marked by the outermost GZW (cf. Shipp et al., 1999; D1, Fig. 3). Between Coulman Island and Drygalski Ice Tongue, a prominent cluster of MSGLs indicates trough-parallel flow (Fig. 3). Therefore, we interpret northward flow at the LGM from at least as far south as the David Glacier outlet (Drygalski Ice Tongue) to a grounding-line north of Coulman Island.

In JOIDES Trough, maximum ice extent is suggested to be recorded by the large-scale GZW (J1) on the mid-outer-shelf (Fig. 3). We base this hypothesis primarily on the presence of up to 8 m of draped glacimarine sediments in the outer trough shown in high-frequency seismic data (Fig. 3a). The observation of LGM-age carbonates on surrounding banks (Taviani et al., 1993) and the presence of LGM-age tephra layers in glacimarine sediments on the outer shelf (Licht et al., 1999) further support this interpretation. Straight furrows that occur seaward of this LGM limit are interpreted as iceberg furrows formed seaward of the LGM grounding line, rather than linear furrows, based on orientations that lack parallel conformity.

The LGM limit in Pennell Trough coincides with the large-scale GZW (P1, Fig. 3), located ~120 km landward of the shelf break (Howat and Domack, 2003). High-frequency seismic data show that this GZW prograded across thick glacimarine sediments that fill the outer trough (Fig. 3b, c).

Large-scale GZWs at the shelf break in each ERS trough (Fig. 3), linear features that extend across the outer shelf, and extensive shelf-edge gullies (Gales et al., 2012) indicate that grounded ice likely reached the shelf break in the ERS (Shipp et al, 1999; Mosola and Anderson 2006). Thin glacimarine sediments on the outer shelf suggest a relatively shorter period of ice-free conditions than in the WRS, and would be consistent with a shelf-break LGM position.

Figure 3 shows the interpreted LGM grounding line and paleo-flow directions derived from the seaward-most linear features that are assumed to represent LGM flow conditions. Generally, linear features delineate trough-parallel flow, which is consistent with previous LGM flow reconstructions (Shipp et al., 1999; Mosola and Anderson, 2006; Anderson et al., 2014).

## 5.2  Western Ross Sea deglaciation

With the exception of Drygalski Trough, the WRS contains sparse and isolated patches of linear features, providing only
glimpses of subglacial flow behaviour and direction despite extensive multibeam bathymetric coverage. Therefore, most paleo-drainage interpretations in the WRS are based on ice-marginal features.

In Drygalski Trough, the ice sheet decoupled from the seafloor and back-stepped rapidly from its LGM position near Coulman Island to a mid-shelf position at Drygalski Ice Tongue, as evidenced by the pristine nature of MSGLs and lack of overprinting ice-marginal landforms. South of Drygalski Ice Tongue, sparse data with poor quality results from the typical
presence of pervasive sea ice. The most prominent deglacial features are a series of intermediate-size GZWs that back-step westward towards Mackay Glacier from a location north of Ross Island (Greenwood et al., 2012).

In JOIDES and Pennell troughs, fields of closely spaced, small-scale GZWs and marginal moraines (Figs. 2d-f) dominate the seafloor, indicating that ice remained in contact with the seafloor during retreat. This implies that overall deglaciation was punctuated by pauses that were long enough to form a small recessional feature, before retreating and forming another
recessional feature. Retreat slowed and the grounding line stabilized in the southernmost part of JOIDES Trough at an intermediate-scale GZW (J2, Fig. 3). A subglacial meltwater channel extending from GZW J2 to the south was likely linked to a large meltwater system that was active during deglaciation. We observe meltwater channels in southern JOIDES and Pennell troughs, which are associated with retreat of the grounding line from positions of stability (J2, and the Pennell Saddle), leading to final rapid deglaciation of grounded ice in the two troughs. The effect of channelized subglacial
meltwater on grounding-line stability is still under investigation.

Ice in the deep Central Basin appears to have retreated quickly, leaving only isolated clusters of recessional moraines. Based on the orientations of these moraines, we interpret a grounding-line embayment that opened over the Central Basin, followed by grounding-line retreat toward the east and west (Fig. 5). Fields of closely spaced, small-scale ice-marginal features in the Central Basin indicate that ice remained in frequent contact with the bed during deglaciation of this area. Because ice did not

lift off from the deep seafloor first, we infer that retreat behaviour was controlled by a steep ice profile rather than physiography, as the ice did not decouple concentrically according to depth contours.

North of Central Basin, extensive fields of small-scale GZWs and moraines record grounding-line retreat onto banks (Figs. 6a-b; 7), indicating that ice remained grounded on WRS banks during deglaciation. The presence of GZWs implies that ice was actively flowing across the banks and mobilizing sediment in order to deposit these marginal features. Thus, WRS banks housed semi-independent ice rises during the late stages of ice sheet retreat from the WRS (Shipp et al., 1999; Anderson et al., 2014; Matsuoka et al., 2015). These findings are supported by modelling results that indicate the presence of independent, detached ice rises on WRS banks late in deglaciation (Golledge et al., 2014). Additionally, Yokoyama et al. (2016) argue that a grounded ice shelf remained pinned on WRS banks until the late Holocene.

## 5.3 Eastern Ross Sea deglaciation

Linear features on the ERS seafloor are overprinted only by large-scale GZWs (Fig. 3). These large-scale GZWs likely record periods of grounding-line stabilization, punctuated by episodes of ice-sheet decoupling and grounding-line retreat that back-stepped tens to hundreds of kilometres in distance and preserved linear features.

We propose two alternative scenarios to explain the observed changes in flow orientation in the ERS. The first scenario ('dynamic flow-switching model') is characterized by alternating regional flow direction throughout the LGM, followed by north-south recession of the grounding line (Fig. 8a). In the second scenario ('embayment scenario'), the ice stream occupying Whales Deep experienced extensive retreat, forming a large grounding-line embayment in the ERS (Fig. 8b).

The dynamic flow-switching scenario requires significant flow reorganization with westward ice flow out of Marie Byrd Land (d1, Fig. 8a) followed by eastward flow across the inter-ice-stream ridge between Whales Deep and Glomar Challenger Basin (d2, Fig. 8a). Trough-parallel flow was then established (d3, Fig. 8a) and ice then began to retreat landward from the continental shelf in all ERS basins, interrupted by phases of grounding-line stabilization and formation of the large GZWs in Whales Deep and Glomar Challenger Basin (d4, Fig. 8a). Different generations of MGSLs are preserved as the grounding-line retreats, but we would not expect them to be preserved if a re-advance or major new episode of streaming had occurred, remoulding the bedform field. Although there have been examples of preserved flow fabrics during events of flow-switching (Stokes et al., 2009; Winsborrow et al., 2012) or at localized patches of basal friction (Stokes et al., 2007; Kleman and Glasser, 2007), the preservation of such extensive flow fabrics throughout three different ice flow configurations is unlikely.

The embayment scenario proposes the formation of an embayment over Whales Deep, based on the presence of large flowsets in surrounding basins that flow across neighbouring inter-ice-stream ridges into Whales Deep (flowsets g, k, Fig. 4). Trough-parallel flow likely occurred first (e1, Fig. 8b), as evidenced by the relatively undisturbed trough-parallel flowset in the outermost part of Whales Deep. During trough-parallel flow, ice grounded on inter-ice-stream ridges was likely sluggish

and strongly coupled to the bed (Klages et al., 2013). An embayment in the Whales Deep grounding line formed (e2, Fig. 8b), drawing flow from outer Glomar Challenger Basin across the inter-ice-stream ridge into Whales Deep and depositing a large-scale GZW on the mid-shelf (W2, Fig. 3). The grounding-line embayment then retreated further towards the Whales Deep inner shelf (e3, Fig. 8b), drawing ice from Glomar Challenger Basin and Little America Basin and prompting flow across the inter-ice-stream ridges into Whales Deep. The ice stream feeding Whales Deep at the LGM may have experienced stagnation or outrun its inner-shelf ice source, destabilizing grounded ice on the outer shelf and causing an embayment to form. Modern Siple Coast ice streams have been observed to slow and stagnate (Anandakrishnan and Alley, 1997; Joughin and Tulaczyk, 2002), suggesting dynamic behaviour in the past.

Shipp et al. (1999) identify the inter-ice-stream ridges in the ERS as aggradational features, meaning that they were centres of focused sedimentation. Embayment grounding lines would have stabilized on the edges of the inter-ice-stream ridges on either side of Whales Deep, transporting sediment to these bathymetric features and aggrading the inter-ice-stream ridges. A large embayment over the ERS is also compatible with the interpreted WRS deglaciation pattern, where a steep EAIS profile is inferred. The formation of an embayment in the ERS is consistent with grounding-line recession in the ERS prior to the WRS (Fig. 6c), followed by east-to-west deglaciation of the WRS.

The two retreat models described here imply a succession of events that can be tested. Greater coverage of high-resolution multibeam data in outer Glomar Challenger Basin, illuminating cross-cutting relationships between flowsets, is crucial for establishing a relative chronology of cross-trough versus trough-parallel flow. Additional multibeam surveys of inter-ice-stream ridges would also provide a better understanding of their role in directing the general flow pattern (cf. Klages et al., 2013). Furthermore, reliable marine radiocarbon dates constraining grounding-line retreat on the Whales Deep inner shelf might provide evidence for early retreat and the formation of a long-lived grounding-line embayment. Based on the available data in this study, the embayment scenario is favoured, due to the landform preservation issues inherent to the dynamic flow-switching model.

The Ross Sea geomorphological record permits us to reconstruct the pattern of ice flow and retreat independently of a radiocarbon chronology and the associated problems therein. Figure 7 presents reconstructed steps in grounding-line retreat that illustrate deglacial patterns across the Ross Sea. Observed grounding lines are linked together to form discrete episodes of deglaciation. These linkages are based on similar morphologies of observed GZWs, extension of grounding line orientations along bathymetric depth contours, and interpretation of local (albeit qualitative) retreat rates based on geomorphic features. Southern Drygalski Trough was the last area in the WRS to experience grounding-line retreat, as outlet glaciers (e.g. Mackay Glacier, and David Glacier flowing into the Drygalski Ice Tongue) receded toward the west and north, leaving fields of moraines and GZWs (Greenwood et al., 2012; Anderson et al., 2014), (Fig. 7). Drainage from the EAIS flowed into the Ross Sea Embayment until the last stage of deglaciation (Fig. 7, steps 7-8). We infer a steep EAIS ice profile

over the WRS throughout deglaciation, based on the contribution of EAIS ice through the Transantarctic Mountains and grounding-line recession unaffected by topography in the central WRS (Fig. 5).

## 5.4 Comparison with existing deglacial models

Currently, there are two very different published retreat scenarios for the Ross Sea (Fig. 9). One of these models, the often
cited 'swinging gate' model (e.g. McKay et al., 2008; Hall et al., 2013), calls for a linear grounding line retreat across the Ross Sea, hinged just north of Roosevelt Island and extending to the Transantarctic Mountains (Conway et al., 1999). This model is constrained by the initiation of ice-divide flow over Roosevelt Island, and two locations along the Transantarctic Mountains with ages from ice-free coastlines, and it indicates deglaciation of the WRS at a faster rate than the ERS. The swinging gate model implies that controls on ice-sheet dynamics were the same throughout the Ross Sea and that
physiography had little influence on ice retreat. This model also implies that ice-sheet retreat from the Ross Sea was controlled mainly by changes in the WAIS catchment, suggesting very high rates of southward retreat along the coast of the Transantarctic Mountains (Conway et al., 1999; Hall et al., 2013, 2015). Alternatively, the 'saloon door' model proposes early retreat in the ERS with a grounding-line embayment in the central Ross Sea (Ackert, 2008). The implied drainage pattern of the saloon door model requires significant inputs from both the EAIS and the WAIS. This model is supported by
cosmogenic exposure ages indicating a thinner ice-sheet profile in the central Ross Sea than at the margins of the WAIS (Parizek and Alley, 2004; Waddington et al., 2005; Anderson et al., 2014).

This study and previous marine studies (Licht et al., 1996; Cunningham et al., 1999; Anderson et al., 2014) suggest early grounding-line retreat within the northern Drygalski Trough, consistent with the swinging gate model. However, reconstructed grounding-line retreat on the remaining Ross Sea continental shelf contrasts with the swinging gate model
(Fig. 7). In particular, our marine-based reconstruction suggests persistent EAIS drainage into the WRS throughout deglaciation, and indicates significant regional variations in grounding-line behaviour between troughs and across banks. Our reconstruction supports the presence of a grounding-line embayment in the ERS, similar to the saloon door model. Here, grounding-line recession in Glomar Challenger Basin is interpreted to precede retreat in the WRS (Fig. 6c), destabilizing grounded ice in southern Pennell Trough and the Central Basin. Deglaciation of the ERS prior to the WRS supports the
observation of the EAIS as a persistent feature in the WRS throughout deglaciation.

Neither the swinging gate nor the saloon door model incorporate observations from the continental shelf, and, as we show here, are not able to fully capture the complexity of grounding-line retreat across the Ross Sea. Our new marine-based model (Fig. 7), reconstructed from comprehensive mapping of seafloor geomorphic features that directly record grounding-line retreat, can now be used to interpret more detailed Ross Sea paleo-ice sheet behaviour and identify regional differences in
deglacial behaviour. Our glacial geomorphic reconstruction independently converges with recent numerical modelling. Model results demonstrate significant EAIS and WAIS contributions to ice flow in the Ross Sea, and suggest that

deglaciation was initiated in Ross Sea troughs and influenced by bedrock highs (Golledge et al., 2014; McKay et al., 2016; DeConto and Pollard, 2016). Additionally, DeConto and Pollard (2016) reproduce an early ERS grounding-line embayment confined to Whales Deep and a WRS Central Basin embayment receding to the east and west, while Golledge et al. (2014) simulate repeated occupation of WRS banks by semi-independent ice rises. Regional reconstructions between models and geologic observations are therefore becoming more and more consistent; however, smaller-scale patterns of grounding-line retreat are not yet reproduced at the resolution of modern numerical models. These localized retreat patterns are important for understanding grounding-line dynamics and smaller-scale processes that drive regional ice behaviour. A key target for further refining such efforts must undoubtedly be a robust and reliable radiocarbon chronology.

## 5.5 Physiographic and geological controls on deglaciation

Many cycles of glacial erosion and deposition have led to Antarctic continental shelves characterized by a fore-deepened shelf profile with exposed bedrock on the inner shelf and thicker sediments on the outer shelf (Anderson, 1999). Runaway grounding-line retreat can occur as ice retreats from the outer to inner shelf due to a lack of pinning points to stabilize the grounding line (e.g. Mercer, 1978; Jamieson et al., 2012). In the WRS, banks and volcanic seamounts provided stable pinning points during deglaciation (Anderson et al., 2014; Simkins et al., in press), and bathymetric highs continue to stabilize the modern Siple Coast ice sheet and ice shelf (Matsuoka et al., 2015). Ice-marginal features are observed to back-step up onto WRS banks (Fig. 6), demonstrating a strong physiographic control on grounding line behaviour. These banks served as pinning points for retreating ice streams and likely evolved into semi-independent ice rises during deglaciation. WRS banks supported an extensive ice shelf that buttressed WRS grounding lines and contributed to the long-lived presence of the EAIS in the WRS (Anderson et al., 2014; Yokoyama et al., 2016). While slight bottlenecking of ERS inter-ice-stream ridges may have played a role in determining positions of grounding-line stability and the formation of large-scale GZWs (Mosola and Anderson, 2006), the ERS seafloor is much more topographically subdued than in the WRS. A lack of high-relief banks and troughs permitted more variable flow in the ERS, but did not allow for pinning and stabilization of ice streams as occurred in the WRS.

In addition to physiography, seafloor substrate has also been argued to exert a fundamental control on ice behaviour, as indicated by variations in geomorphic features across different substrates (e.g. Wellner et al., 2001; Larter et al., 2009; Graham et al., 2009). In Antarctica, studies have shown that ice streams flowing across soft, deformable sedimentary beds are characterized by MSGLs (e.g. Wellner et al., 2001, 2006; Ó Cofaigh et al., 2002, 2005; Graham et al., 2009). Ice flowing over unconsolidated beds can mobilize subglacial sediments and develop a thick layer of pervasive deformation till, facilitating faster ice flow than is possible by internal ice deformation (Alley et al., 1989). By contrast, crystalline bedrock or older and more consolidated strata outcropping on the seafloor are more resistant to glacial erosion, preventing the development of deforming till underneath a flowing ice stream, and are associated with bedrock erosional features such as drumlinoids that indicate slower ice-flow velocities and stick-slip motion. An excellent example is the field of drumlinoids in

inner Glomar Challenger Basin that corresponds to a localized area of outcropping bedrock (Figs. 2c, Fig. 10). At the point where sedimentary deposits lap onto bedrock, these drumlinoids transition seaward into MSGLs (Anderson, 1999).

The compilation of geological data in Fig. 10 shows the strata beneath the most recent observable glacial erosional surface, representing the substrate that ice flowing across the continental shelf at the LGM would have encountered. The degree of consolidation of these strata is derived from information obtained from drill cores collected during Deep Sea Drilling Project Leg 28, extrapolated to high-resolution seismic stratigraphic correlations across the Ross Sea (Anderson and Bartek, 1992; Alonso et al., 1992; Anderson, 1999; Bart et al., 2000). The WRS is characterized by more variable geology and by older substrate, while mostly unconsolidated Plio-Pleistocene sediments blanket the ERS shelf. Thick and extensive unconsolidated sediments likely contributed to a pervasive layer of deformation till in the ERS (Mosola and Anderson, 2006). This thick layer of deformation till facilitated fast flowing ice and transported sediment to large-scale GZWs through a classic till conveyor-belt mechanism. Fast-flowing ice likely contributed to a low-profile ice sheet that episodically decoupled from the seafloor during retreat from the continental shelf (Mosola and Anderson, 2006). More consolidated strata outcropping in the WRS may have limited such pervasive subglacial deformation, potentially causing slower ice stream velocities in WRS troughs. This characteristic seafloor geology, coupled with numerous pinning points, was conducive to a higher profile ice sheet that remained in contact with the seafloor throughout much of its retreat from the continental shelf.

Grounded ice in Little America Basin flowed over its eastern bank and converged with an outlet glacier draining Marie Byrd Land (flowset *n*, Fig. 4). This flow pattern implies that at one point, Little America Basin was not able to drain all of the ice flowing into it and therefore some of that ice was forced eastward out of the trough. During the LGM, Little America Basin ice streams flowed across late Oligocene and Miocene sedimentary rocks (Fig. 10). Thus, it was more resistant to ductile subglacial deformation than the substrates encountered by other ice streams flowing across the ERS. Resulting flow velocities were therefore not high enough to transport all of the ice entering the Little America Basin outlet, some of which was captured and funnelled into the neighbouring outlet.

Physiography exerts a first-order control on regional ice stream flow and retreat dynamics, and seafloor geology plays an important subsidiary role in controlling ice behaviour. These controls influence regional retreat patterns; more localised ice behaviour is still under investigation. Numerous other processes affect glacial dynamics, such as ice-shelf buttressing, sediment shear strength and ice-bed coupling, and subglacial meltwater (e.g. Boulton et al., 2001; Dupont and Alley, 2005; Stearns et al., 2008). External forcings such as tidal effects, circumpolar deep water incursion and under-melting of ice shelves, and atmospheric effects are also influential (e.g. Rignot, 1998; Zwally et al., 2002; Arneborg et al., 2012; Walker et al., 2013). Ross Sea retreat was asynchronous between troughs, suggesting differential responses to these processes. Ongoing work on characterizing Ross Sea glacial geomorphology highlights the effect of these forcings on local grounding-line stability.

**6 Conclusions**

During the LGM, grounded ice reached the continental shelf break in the ERS, but not in the WRS. The WRS seafloor is characterized by geomorphic features that indicate periods of rapid recession following the LGM, and record the persistent presence of a steep-profiled EAIS in the WRS throughout deglaciation. Retreat in the ERS was likely initiated by the formation of a large grounding-line embayment across Whales Deep. Based on the interpretation of glacial geomorphic features, Glomar Challenger Basin in the ERS is believed to have been completely deglaciated prior to retreat of grounded ice from the deep Central Basin in the WRS.

Considering the complex glacial geomorphic assemblages across the entire Ross Sea shelf, the 'swinging gate' and 'saloon door' models both fail to fully capture the style of deglaciation. The saloon door model is more consistent with glacial geomorphic observations on the Ross Sea continental shelf, describing a mode of deglaciation that may have occurred in more than one sector as ice retreated into its component sub-catchments. Based on this study, we conclude that it is eminently clear that deglaciation across the Ross Sea shelf did not involve a linear grounding line across the multiple troughs and banks. Additional analyses of Ross Sea continental shelf sedimentology and additional reliable radiocarbon ages marking grounding-line retreat are necessary to test and refine the deglacial patterns proposed here. A radiocarbon chronology will help integrate our grounding-line reconstruction with previous work done on Ross Sea deglacial history.

Major differences between regional retreat characteristics are attributed to physiography. Ice was pinned on the high-relief banks in the WRS, whereas the lack of comparable features in the ERS indicates that the WAIS was not stabilized by pinning points. Similar physiographic controls are likely buttressing the modern Siple Coast grounding line. Seafloor geology played a secondary role in influencing paleodrainage patterns. Younger and relatively unconsolidated Plio-Pleistocene sediments in the ERS, with the exception of Little America Trough, are associated with fast ice flow, whereas the older and more consolidated strata that characterized the WRS seafloor may have hindered pervasive till deformation and contributed to slower ice-stream velocities. These observations can be generalized to other locations with regional seafloor geologic variation, such as the Weddell Sea Embayment. The controls on flow behaviour and retreat patterns revealed in our new Ross Sea deglacial reconstruction can now be incorporated into future work on understanding marine ice-sheet behaviour at the modern grounding line and across the Antarctic continental shelf.

**Acknowledgements**

This research was funded by the National Science Foundation (NSF-PLR 1246353 to J.B.A) and the Swedish Research Council (D0567301 to S.L.G.). The authors thank the crew of the RV/IB *Nathaniel B. Palmer* and Antarctic Support Contract staff for a successful cruise. We thank Kathleen Gavahan for providing assistance with multibeam datasets, students

from Rice University, University of Houston and Louisiana State University for participating in data collection and processing, and Jean Aroom and the Fondren GIS Center, Rice University, for technical support.

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

Supplementary Table 1. Multibeam dataset compilation. NBP1502 cruise data were used here with the permission of J.B. Anderson. The remaining Nathaniel B. Palmer (NBP) cruise data were accessed through the Lamont-Doherty Earth Observatory (*marine-geo.org*) and the Oden data are available at *oden.geo.su.se*. Swedish Polar Research = SPR.

| Cruise Number | Vessel | Multibeam System | Date | PI |
|---|---|---|---|---|
| NBP9801 | NBP | SeaBeam Instruments 2112 | 1/16/1998 - 2/18/1998 | J.B. Anderson |
| NBP9802 | NBP | SeaBeam Instruments 2112 | 2/22/1998 - 4/2/1998 | S. Honjo |
| NBP9803 | NBP | SeaBeam Instruments 2112 | 5/1/1998 - 6/17/1998 | M. Jeffries/D. Garrison |
| NBP9807 | NBP | SeaBeam Instruments 2112 | 11/1/1998 - 12/12/1998 | R. Dunbar |
| NBP9901 | NBP | SeaBeam Instruments 2112 | 12/26/1998 - 2/4/1999 | M. Jeffries |
| NBP9902 | NBP | SeaBeam Instruments 2112 | 2/12/1999 - 3/22/1999 | J.B. Anderson |
| NBP9909 | NBP | SeaBeam Instruments 2112 | 12/20/1999 - 2/9/2000 | J. Bengtson |
| NBP0001 | NBP | SeaBeam Instruments 2112 | 2/14/2000 - 3/30/2000 | S. Jacobs/T. Kellogg |
| NBP0209 | NBP | Kongsberg EM120 | 12/11/2002 - 12/30/2002 | S. Cande |
| NBP0301 | NBP | Kongsberg EM120 | 1/5/2003 - 1/29/2003 | L. Bartek/B. Luyendyk |
| NBP0301A | NBP | Kongsberg EM120 | 2/1/2003 - 2/18/2003 | P. Bart |
| NBP0301B | NBP | Kongsberg EM120 | 2/20/2003 - 2/22/2003 | W. Smith/V. Asper |
| NBP0302 | NBP | Kongsberg EM120 | 2/24/2003 - 4/4/2003 | A. Gordon |
| NBP0305A | NBP | Kongsberg EM120 | 12/20/2003 - 12/30/2003 | W. Smith |
| NBP0306 | NBP | Kongsberg EM120 | 1/4/2004 - 1/15/2004 | B. Luyendyk/L. Bartek |
| NBP0401 | NBP | Kongsberg EM120 | 1/19/2004 - 2/17/2004 | T. Wilson |
| NBP0402 | NBP | Kongsberg EM120 | 2/21/2004 - 4/6/2004 | M. Visbeck |
| NBP0408 | NBP | Kongsberg EM120 | 10/12/2004 - 12/6/2004 | S. Jacobs |
| NBP0409 | NBP | Kongsberg EM120 | 12/18/2004 - 1/21/2005 | R. Kiene/D. Kieber |
| NBP0501 | NBP | Kongsberg EM120 | 1/28/2005 - 2/13/2005 | A. Gordon |
| NBP0508 | NBP | Kongsberg EM120 | 10/26/2005 - 12/3/2005 | P. Neale |
| NBP0601 | NBP | Kongsberg EM120 | 12/17/2005 - 1/24/2006 | G. DiTullio |
| NBP0601A | NBP | Kongsberg EM120 | 1/30/2006 - 2/2/2006 | W. Smith |
| NBP0602 | NBP | Kongsberg EM120 | 1/30/2006 - 2/21/2006 | J. Stock |
| NBP0608 | NBP | Kongsberg EM120 | 11/3/2006 - 12/11/2006 | G. DiTullio |
| NBP0701 | NBP | Kongsberg EM120 | 12/22/2006 - 1/28/2007 | S. Cande/P. Castillo |
| NBP0702 | NBP | Kongsberg EM120 | 2/2/2007 - 3/23/2007 | S. Jacobs |
| OSO0708 | Oden | Kongsberg EM122 | 11/29/2007 - 1/7/2008 | SPR Secretariat |
| NBP0801 | NBP | Kongsberg EM120 | 1/9/2008 - 1/26/2008 | D. Caron/B. Huber |
| NBP0802 | NBP | Kongsberg EM120 | 1/30/2008 - 2/20/2008 | D. Caron/P. Bart |
| NBP0803 | NBP | Kongsberg EM120 | 2/22/2008 - 3/13/2008 | P. Bart |
| NBP1005A | NBP | Kongsberg EM120 | 1/13/2010 - 1/16/2011 | P. Yager |
| OSO0910 | Oden | Kongsberg EM122 | 2/8/2010 - 3/12/2010 | M. Jakobsson/J.B. Anderson |
| NBP1005 | NBP | Kongsberg EM120 | 11/26/2010 - 1/16/2011 | P. Yager |
| OSO1011 | Oden | Kongsberg EM122 | 12/8/2010 - 1/16/2011 | SPR Secretariat |
| NBP1101 | NBP | Kongsberg EM120 | 1/19/2011 - 2/15/2011 | J. Kohut/A. Kutska |
| NBP1102 | NBP | Kongsberg EM120 | 2/19/2011 - 4/23/2011 | J. Swift |
| NBP1201 | NBP | Kongsberg EM120 | 12/24/2011 - 2/11/2012 | D. McGillicuddy |
| NBP1202 | NBP | Kongsberg EM120 | 2/11/2012 - 2/27/2012 | H. Owen |
| NBP1210 | NBP | Kongsberg EM120 | 1/6/2013 - 2/9/2013 | K. Halanych |
| NBP1302 | NBP | Kongsberg EM120 | 2/12/2013 - 4/5/2013 | D. Hansell/X. Yuan/G. Kooyman |
| NBP1310B | NBP | Kongsberg EM120 | 12/3/2013 - 1/23/2014 | K. Arrigo/R. Aronson |
| NBP1502A | NBP | Kongsberg EM122 | 1/23/2015 - 3/20/2015 | J.B. Anderson |

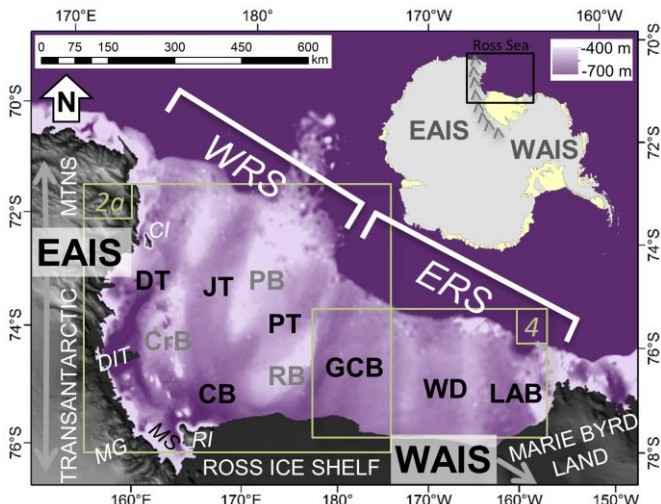

Figure 1. Regional bathymetry of the Ross Sea continental shelf, with bed topography data acquired from BEDMAP2 (Fretwell et al., 2013). Inset shows the West and East Antarctic ice sheets (WAIS and EAIS, respectively), separated by the Transantarctic Mountains with the Ross Sea study area outlined. Locations for Fig. 2a and Fig. 4 are shown. WRS (Western Ross Sea), ERS (Eastern Ross Sea), EAIS (East Antarctic Ice Sheet), WAIS (West Antarctic Ice Sheet), DT (Drygalski Trough), JT (JOIDES Trough), PT (Pennell Trough), CB (Central Basin), CrB (Crary Bank), PB (Pennell Bank), RB (Ross Bank), GCB (Glomar Challenger Basin), WD (Whales Deep), LAB (Little America Basin), CI (Coulman Island), DIT (Drygalski Ice Tongue), MG (Mackay Glacier), MS (McMurdo Sound).

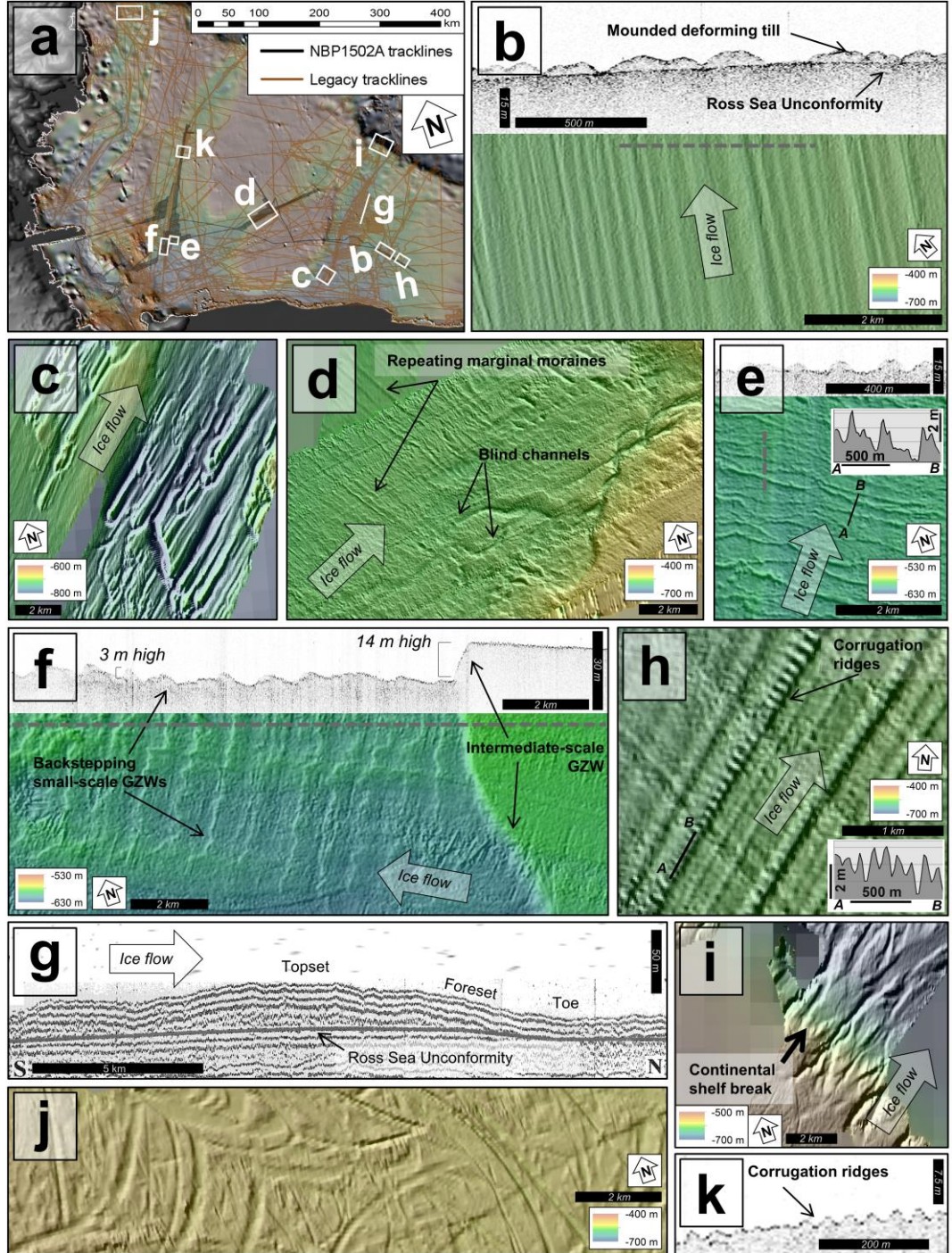

Figure 2. Glacial geomorphic features of the continental shelf. (a) Ross Sea tracklines from cruise NBP1502A (black lines) and legacy (brown) cruises and locations of (b-k). (b) MSGLs (3-5 m amplitude) on the inner shelf of Glomar Challenger Basin occur above a glacial erosional surface, imaged by high-frequency seismic data. (c) Drumlinoids on the inner shelf of Glomar Challenger Basin. (d) A subglacial meltwater channel in Pennell Trough with complex channel morphology,

associated with small-scale recessional ice-marginal features. (e) Marginal moraines in JOIDES Trough. (f) Small-scale and intermediate-scale GZWs in JOIDES Trough. (g) Seismic profile showing GZW (4b) in Glomar Challenger Basin modified from Mosola and Anderson (2006). (h) Linear iceberg furrows with average depth of 14 m; corrugation ridges inside the furrows have heights of 0.5-2 m. (i) Shelf-edge gullies on the eastern Ross Sea continental shelf break. (j) Arcuate cross-cutting iceberg furrows on the outer shelf of Drygalski Trough. (k) Corrugation ridges in outer JOIDES trough, with heights ranging from 0.5-2 m. Dashed lines on the multibeam images indicates the location of the CHIRP profiles (vertical scales were calculated from two-way travel time using the sound velocity conversion of 1500 m/s).

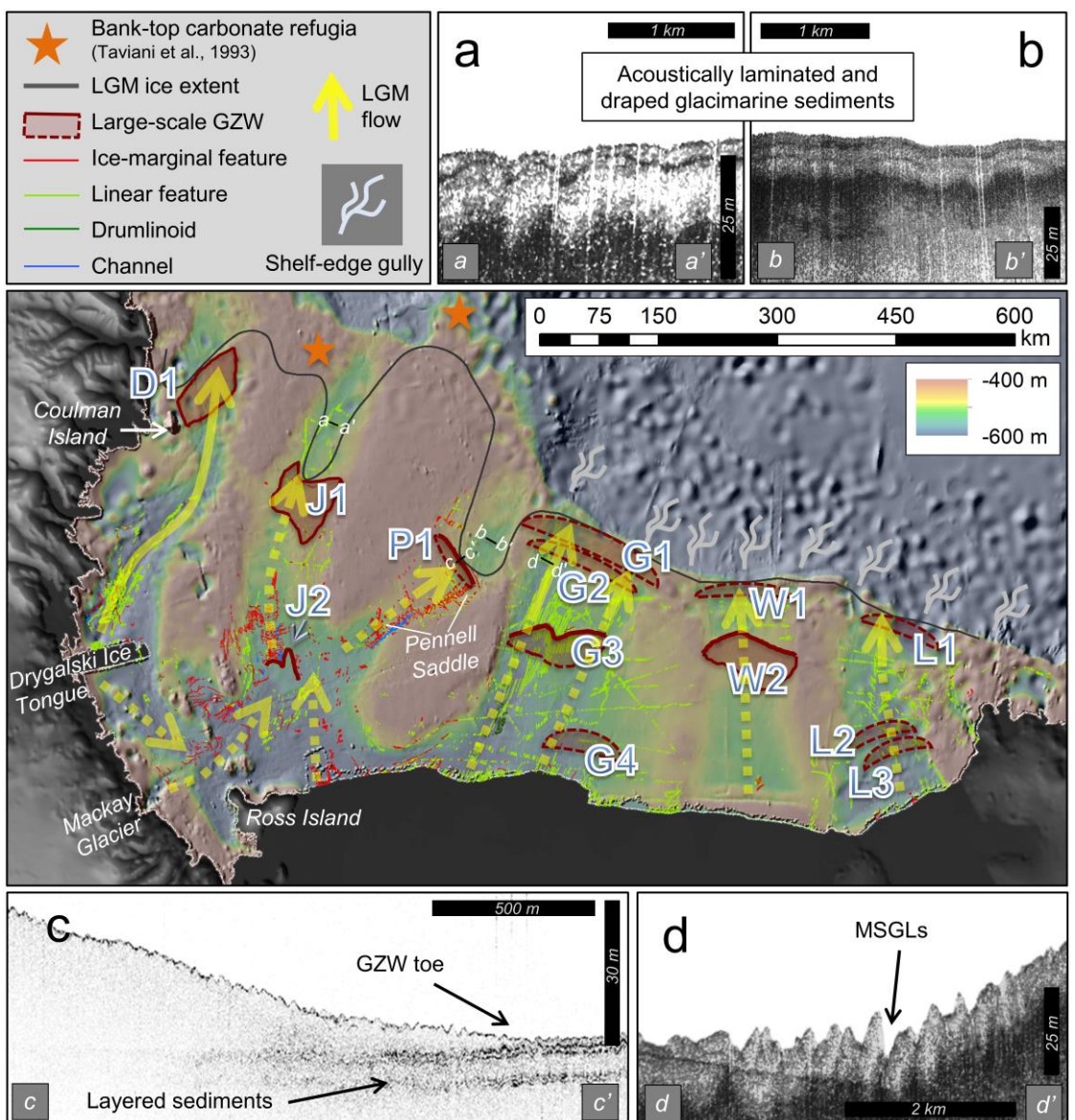

Figure 3. Distribution of geomorphic features and evidence for LGM extent. Large-scale GZWs are outlined with a solid line where the GZW boundary is known, and a dotted line where the boundary is inferred based on depth contours. GZWs that are only identified in seismic lines are symbolized with a dotted lens shape. LGM flowlines based on geomorphic flow indicators are displayed as thick yellow arrows. Dotted arrows in southwestern Ross Sea denote flow patterns based on a geomorphic record of local ice flow out of EAIS outlet glaciers during deglaciation. It remains uncertain whether those flow patterns were also active during the LGM. In the eastern Ross Sea, lineations corresponding to LGM flow are also unclear; we assume that LGM ice streams flowed roughly parallel to trough axes, based on the most seaward flowsets. High-frequency seismic profiles show thick, draped glacimarine sediments in (a) JOIDES Trough (4-8 m thick) and (b) Pennell Trough (9-14 m thick). (c) The LGM GZW foreset in Pennell Trough prograded over thick pre-LGM glacial marine sediments. (d) MSGLs have no appreciable post-glacial sediments in outer Glomar Challenger Basin.

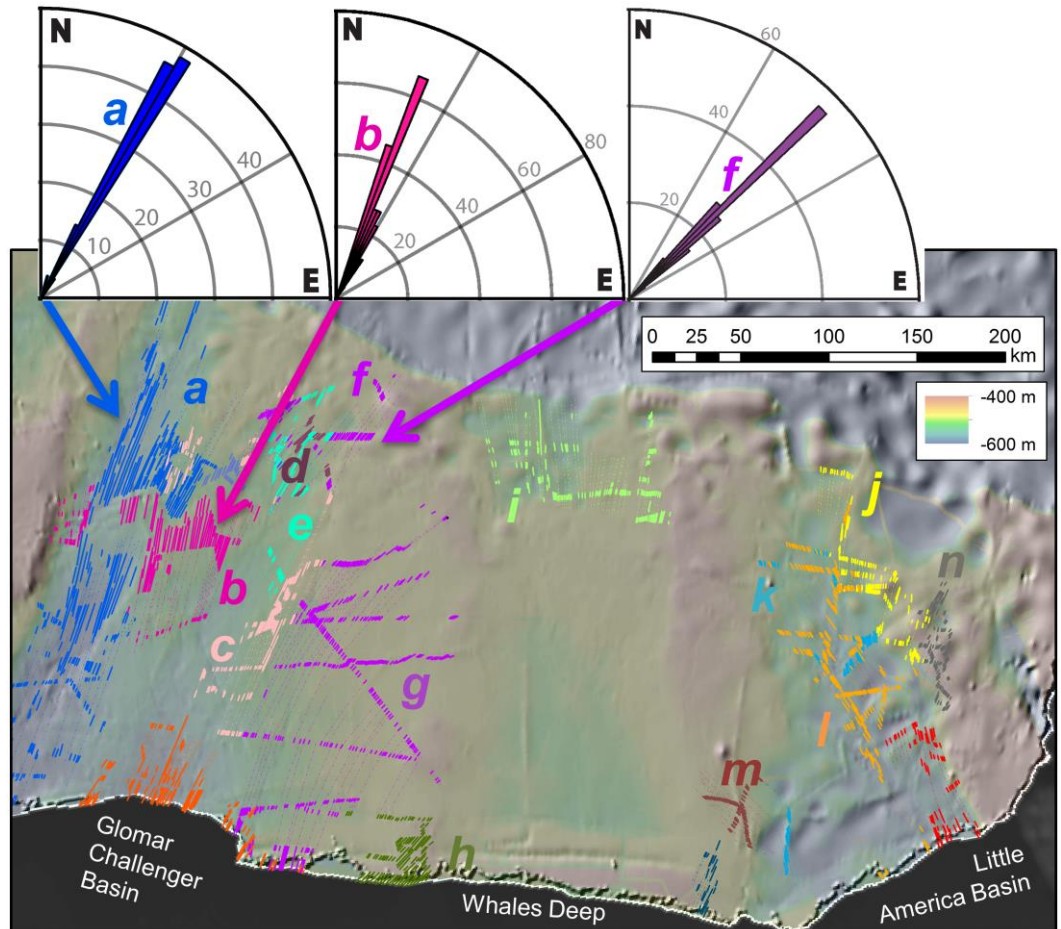

Figure 4. Linear features in the eastern Ross Sea were grouped into flowsets based on features mapped using multibeam data (solid lines) and interpolation between multibeam data (dotted lines). Major flowsets are labeled for reference in text. Flowsets were placed in a relative chronology partially based on maximum seaward extent; each orientation represents a different vintage of flow. Three example flowsets and corresponding Rose diagrams are shown from outer Glomar Challenger Basin.

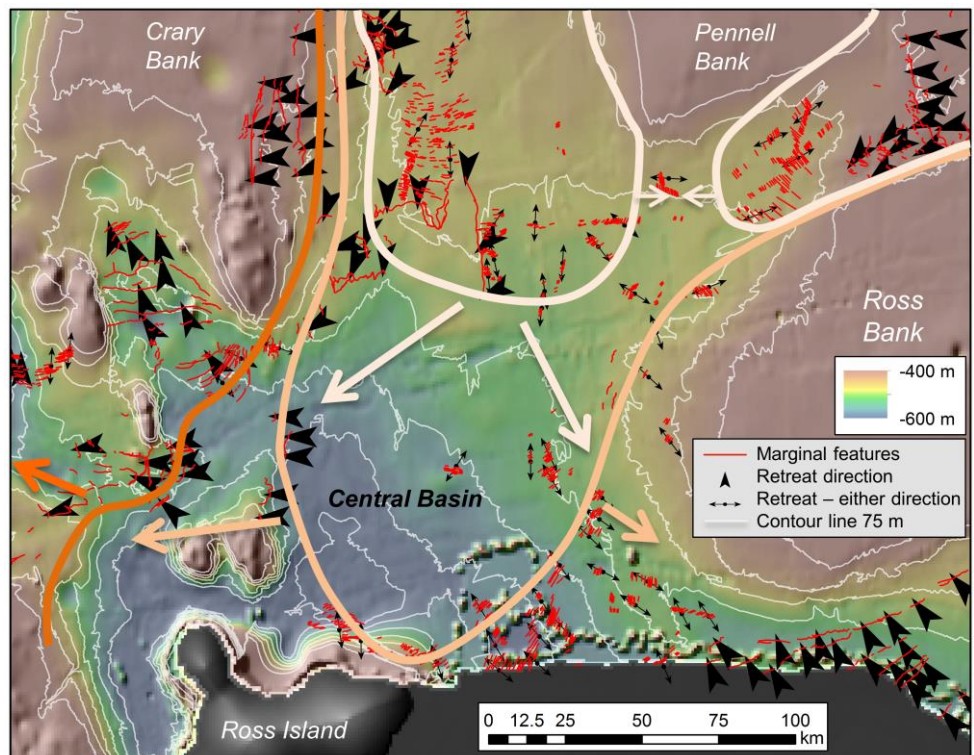

Figure 5. Retreat direction in the western Ross Sea is inferred from GZWs (arrowheads) and symmetric marginal moraines (double-sided arrows). Reconstructed grounding lines (solid lines) are accompanied by large arrows indicating regional retreat. Thin white lines are depth contours at 75 m increments. Deglaciation in the deep Central Basin did not follow depth contours, implying a steep deglacial EAIS ice profile in order for ice to remain grounded across a range of depths contemporaneously.

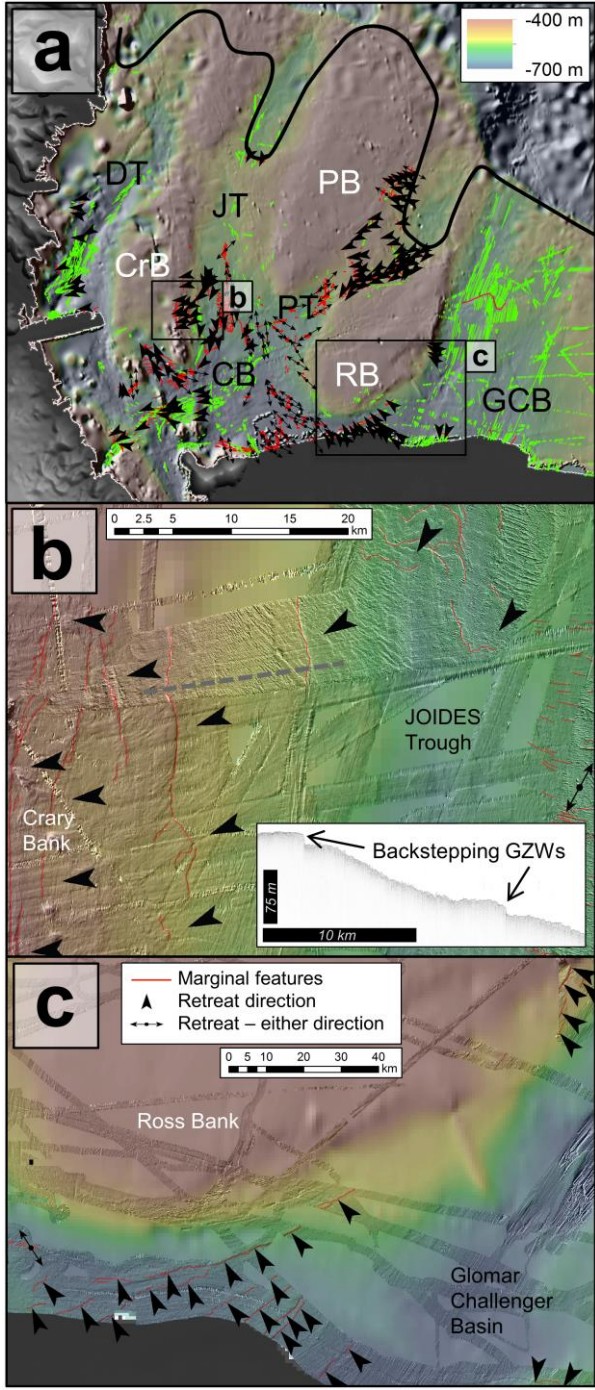

Figure 6. (a-b) Grounding lines are observed to retreat up onto banks, as shown by back-stepping wedges and marginal moraines. Arrowheads denote retreat direction. (c) Back-stepping grounding lines in southwestern Glomar Challenger Basin imply that ice had decoupled there before retreating westward into the WRS. The color scale is consistent between all panels.

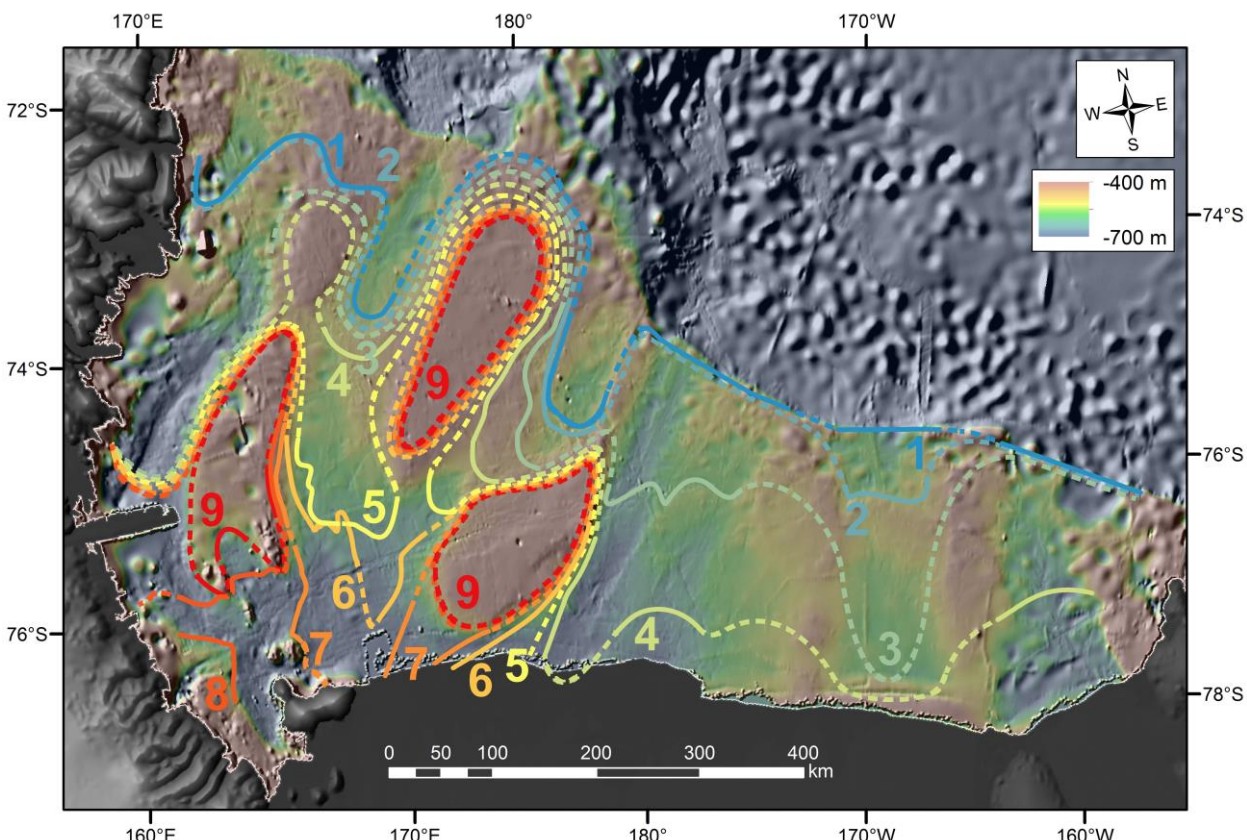

Figure 7. Reconstructed grounding-line retreat across the Ross Sea based on geomorphic indicators of grounding lines (solid lines) and inferred grounding-line locations (dashed). Each line marks a relative step in grounding-line retreat starting with step 1 at the LGM grounding line and ending with step 9 with ice pinned on banks.

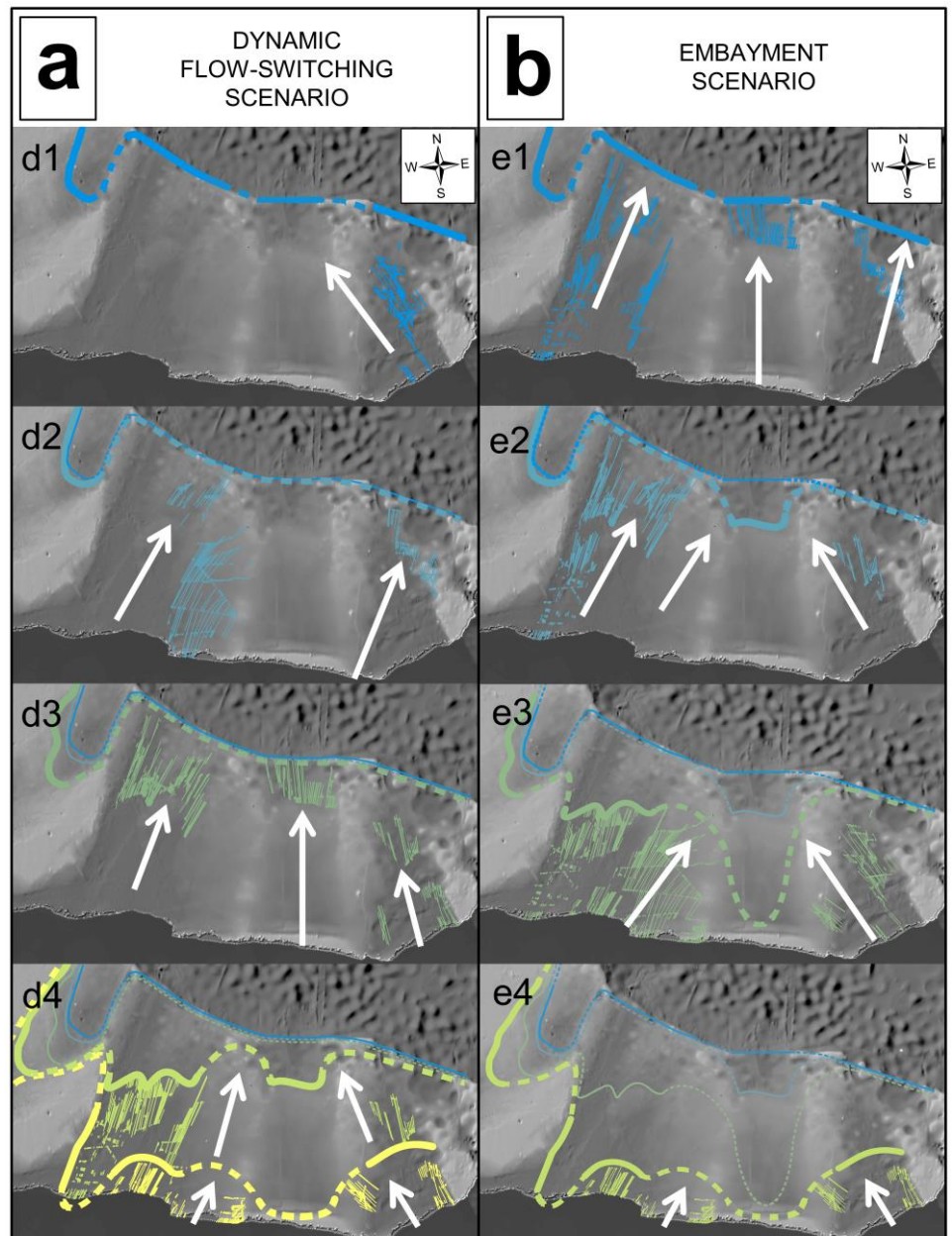

Figure 8. Possible retreat scenarios for the eastern Ross Sea interpreted from flowsets. (a) The dynamic flow-switching scenario calls for alternating regional flow directions, followed by north-south recession of the grounding line. This model requires preservation of at least three different flow fabrics as ice remains grounded on the outer continental shelf. (b) In the embayment scenario, a large grounding-line embayment in the eastern Ross Sea forms over Whales Deep. The embayment scenario is independently more consistent with inland paleo-ice thickness reconstructions and seafloor seismic observations.

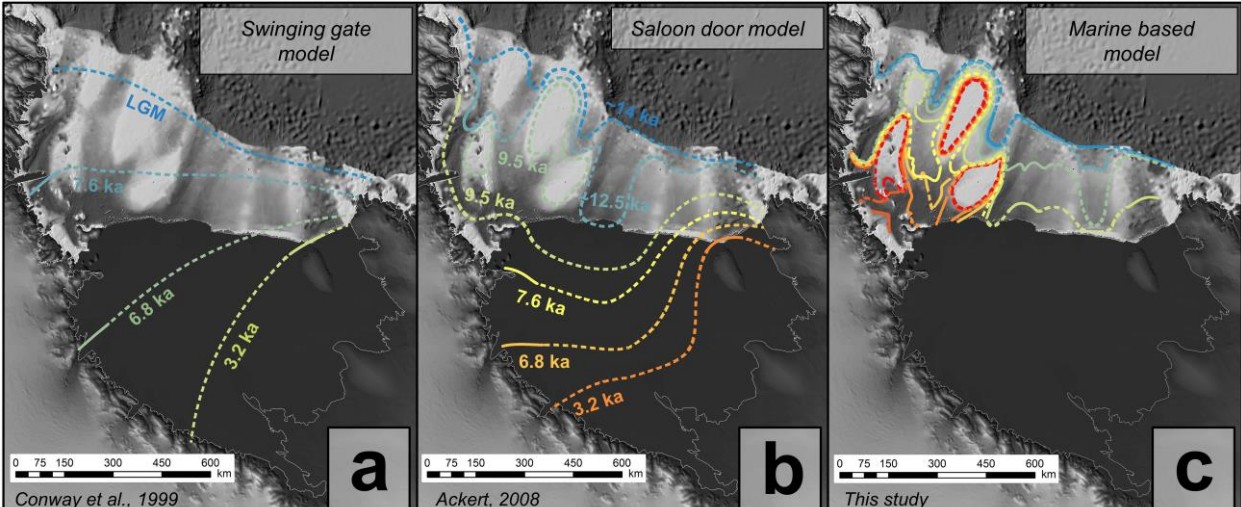

Figure 9. Comparison of existing models of Ross Sea deglaciation. (a) The 'swinging gate model' (Conway et al., 1999) assumes a linear grounding line swinging across the Ross Sea, implying that controls on ice-sheet dynamics are the same throughout the Ross Sea and that physiography has little influence on ice retreat. This model indicates deglaciation of the WRS prior to the ERS, and implies that the Ross Sea was filled with WAIS ice during LGM and throughout deglaciation. (b) The 'saloon door' model of deglaciation suggests early retreat in the ERS with a potential grounding-line embayment in the central Ross Sea (Ackert, 2008), requiring significant inputs from both the EAIS and the WAIS. (c) The marine-based reconstruction presented here uses glacial geomorphology to interpret paleo-grounding-line retreat.

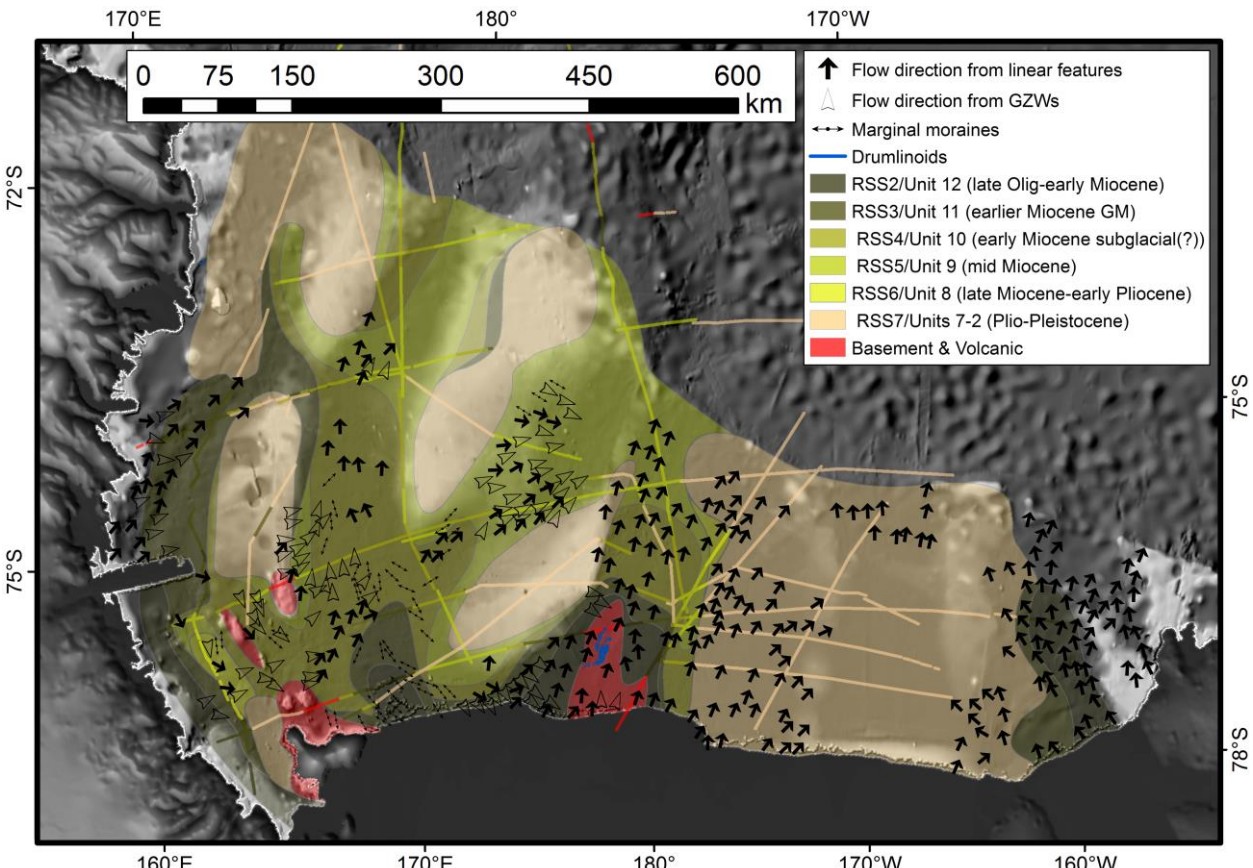

Figure 10. Control of seafloor geology on ice dynamics. Geologic boundaries were interpolated from legacy seismic lines (shown here) with pre-interpreted seismic units by Anderson and Bartek (1992) and Brancolini et al. (1995). The WRS is characterized by complex, older and more consolidated strata, where ice streams have eroded down to Oligocene-age strata. Volcanic islands and seamounts outcrop in the southern portion of the WRS. The western side of Glomar Challenger Basin, bordering Ross Bank, contains older and more variable geologic strata outcropping at the seafloor, including a patch of basement outcrop on the inner shelf. In general, thick unconsolidated Plio-Pleistocene strata fill most of the ERS and increase in thickness in an offshore direction (Alonso et al., 1992). Plio-Pleistocene sediments are thin in southern Whales Deep, overlying older Miocene strata. Eastern Little America Basin is characterized by lithified late Oligocene through Miocene deposits. Arrows indicating flow direction are based on geomorphic features.