# Peer review of "Retreat Scenarios and Changing Controls in the Ross Sea, Antarctica"

_The Cryosphere, 2016_

## Referee Comment (RC1) · N.R. Golledge (Referee) · 8 Mar 2016

Halberstadt et al - Cryosphere

The submitted manuscript presents new and legacy marine geophysical data from the Ross Sea, Antarctica, and uses this to reconstruct the pattern of flow both at the LGM and during its retreat. The paper is very well-written and well illustrated, with clear conclusions that are robust with respect to the data presented. I have no problem in recommending that the paper be accepted with only a few minor edits.

My only gripe really is that there has been quite a lot of Ross Sea work published recently, and not all of it is acknowledged here. This is unfortunate, because the papers

I'm thinking of lend considerable support to the interpretations presented by the present authors and so would nicely bolster their arguments. In particular it would be good to acknowledge McKay et al., 2016 (Geology), who came to very similar conclusions based on different data.

Clearly I have a bias in this regard, but I think it would be good if the efforts of the modelling community were also acknowledged. It is often stated in introductions to 'empirical' studies that the new data will help 'constrain numerical models', indeed, the current authors do this in the very first sentence of the Abstract. But what is the point of modellers using these geological data, if the models they produce are then disregarded? Maybe sometimes the modelling can help with the geological interpretations, rather than the other way around.

There are of course many modelling papers out there, but I know for a fact that Golledge et al., 2012, 2013, and 2014 all mention that retreat most likely started first in the deeper parts of the outer Ross Sea, and that the pattern of retreat was a product of incoming fluxes from both EAIS and WAIS, and was highly dependent on the location of bedrock highs. To illustrate my point, I'm uploading a figure showing the modelled grounding-line positions from the simulations published in McKay et al 2016, overlain on Figure 7 of the submitted paper. Personally I see a considerable amount of agreement there, which is gratifying because it means the models are getting something right!

Anyway, I'm not insisting that the authors have to cite all these papers, but it would be nice to 'close the loop' in a sense and recognise that sometimes synergies between modellers and empiricists can allow a convergence of views that together really show how flawed the 'swinging gate' model is.

Other than this, I can't really find fault with the paper, so I commend the authors for doing a great job pulling the data together and hope to see this published soon.

N R Golledge 8th March 2016

[Figure]

**Fig. 1.** Modelled GLs overlain of Fig. 7

---

## Referee Comment (RC2) · J. P. Klages (Referee) · 9 Mar 2016

Halberstadt et al. combined new multibeam swath-bathymetric data with already existing bathymetric and seismic datasets to present an extensive and comprehensive view of ice sheet extent and retreat in the Ross Sea Embayment (RSE), Antarctica. On the basis of this new compilation the authors were able to reconstruct flow pathways and retreat dynamics during and subsequent to the Last Glacial Maximum across the entire RSE, which led to some new conclusions about the ice sheet history in that region. The paper is well written, easy to grasp, but sometimes slightly lengthy and repetitive. The figures support the text sufficiently, however some figures could be easily combined with others in order to provide more clarity, and to save space. Generally and after

consideration of the edits suggested below, I would like to see this manuscript published as it provides a new and valuable combination of datasets that allow a detailed insight into the Ross Sea Embayment glacial history.

As the editor already pointed out, I would like to see a more detailed implementation of this work with previous work from the area, especially in the introduction. In particular, more recent papers such as Bart and Owolana (2012), QSR and McKay et al. (2016), GEOLOGY need to be considered in this regard. Extensive work has been performed in the RSE and the authors should point out more clearly what is known so far, how their new results fit into these previous results, and how their newly presented results complement and maybe change them. I further encourage the authors to incorporate the results of previous modelling efforts in more detail (e.g. recent studies by Golledge et al.) in order to define synergies. This would reveal the progress already achieved, but would also highlight the need for necessary future work. Building onto that it should be emphasized how empirical future work in the area could focus in order to reduce existing data-model mismatches. The authors should further point out that sediments and reliable radiocarbon dates for min. GL retreat are urgently needed in order to verify their hypotheses. Since their interpretations are exclusively based on geophysical data, they should phrase much more carefully in many parts of the manuscript. The lack of age control should be the strongest motivation for future work in the area. Lastly, the introduction should emphasize the significance of the RSE for Antarctic ice sheet stability in more detail, maybe also in regard to other large embayments such as the Weddell and Amundsen Sea Embayment. Therefore, the significant contributions by The RAISED Consortium (2014) should be incorporated. And – but this is just my personal opinion – I would suggest to slightly change the title of the manuscript to "Past Ice-Sheet Behaviour in the Ross Sea Embayment, Antarctica: Retreat Scenarios and Changing Controls".

If the aforementioned issues and the minor edits and suggestions in the supplementary file will be met sufficiently, I fully support the publication of this manuscript in

"The Cryosphere". Technical corrections, suggestions for improving the readability, and some concerns from my side are listed in a supplementary file by page and line number.

P1, line 8: Replace "on" with "for" (*for* numerical ice-sheet models)

P1, line 12: Delete "in contact with the bed".

P1, line 22: Change to "The Ross Sea Embayment (RSE) drains ~25% of the AIS into the Ross Sea and thus is the largest drainage basin in Antarctica, fed by multiple ice streams...".

P2, line 11: Change to "Multibeam swath bathymetry provides a record of bed conditions beneath the former ice sheet, ...".

P2, line 12: Change to "These landforms record flow behaviour and past thermal regimes of formerly grounded ice."

P2, lines 15-16: Change to "This unique and integrated dataset ... much higher resolution, thereby revealing the palaeo-ice sheet bed with a much higher resolution compared to their modern counterparts."

P2, lines 17-19: Change to "...this dataset to define glacial geomorphic features that characterize past flow and retreat dynamics, thus reconstruct ice-sheet paleodrainage across the Ross Sea Embayment during and subsequent to the LGM."

P2, line 21-22: "Change to "..., which preferentially eroded along pre-existing tectonic lineaments (you may give a reference here)."

P3, line 5: Write "Austral summer".

P3, line 16: Replace "cannibalized" with "eroded" or "obliterated".

P3, line 24: Replace "post-LGM" with "postglacial", since some features may be covered by sediments that started to deposit prior to the LGM.

P3, lines 24-25: Replace "(post-)LGM" with "glacial" since some of the subglacial features in the RSE do not necessarily record LGM ice cover.

Results section:

I suggest renaming section to "Results and interpretation". Descriptions of features and references to similar, already described features elsewhere are largely missing – at least for the new dataset. Which landforms did you detect, how would you describe them, do they resemble already published features, and how do you interpret them on that basis (Description – Reference – Interpretation – Significance).

P3, lines 28-29: Change to "Subglacial landforms form beneath permanently grounded ice that exerts the offset buoyant forces by the ocean."

P4, line 27: Replace "equivocal" with "controversial" and give reference(s) for this statement.

P5, lines 4-5: Rephrase to "Ice-marginal features form within the grounding zone, the transition from permanently grounded ice to ice that decoupled from its bed to become a floating ice shelf."

P5, line 5: Either mark listed features as examples (e.g. GZWs, marginal moraines, ...) or list all of them.

P5, lines 11-12: Not exclusively – large GZWs may also indicate higher sediment flux.

**Fig. 1.**

---

## Referee Comment (RC3) · J. P. Klages (Referee) · 9 Mar 2016

P1, line 8: Replace "on" with "for" (*for* numerical ice-sheet models)

P1, line 12: Delete "in contact with the bed".

P1, line 22: Change to "The Ross Sea Embayment (RSE) drains ~25% of the AIS into the Ross Sea and thus is the largest drainage basin in Antarctica, fed by multiple ice streams…".

P2, line 11: Change to "Multibeam swath bathymetry provides a record of bed conditions beneath the former ice sheet, …".

P2, line 12: Change to "These landforms record flow behaviour and past thermal regimes of formerly grounded ice."

P2, lines 15-16: Change to "This unique and integrated dataset … much higher resolution, thereby revealing the palaeo-ice sheet bed with a much higher resolution compared to their modern counterparts."

P2, lines 17-19: Change to "…this dataset to define glacial geomorphic features that characterize past flow and retreat dynamics, thus reconstruct ice-sheet paleodrainage across the Ross Sea Embayment during and subsequent to the LGM."

P2, line 21-22: "Change to "…, which preferentially eroded along pre-existing tectonic lineaments (you may give a reference here)."

P3, line 5: Write "Austral summer".

P3, line 16: Replace "cannibalized" with "eroded" or "obliterated".

P3, line 24: Replace "post-LGM" with "postglacial", since some features may be covered by sediments that started to deposit prior to the LGM.

P3, lines 24-25: Replace "(post-)LGM" with "glacial" since some of the subglacial features in the RSE do not necessarily record LGM ice cover.

Results section:

I suggest renaming section to "Results and interpretation". Descriptions of features and references to similar, already described features elsewhere are largely missing – at least for the new dataset. Which landforms did you detect, how would you describe them, do they resemble already published features, and how do you interpret them on that basis (Description – Reference – Interpretation – Significance).

P3, lines 28-29: Change to "Subglacial landforms form beneath permanently grounded ice that exerts the offset buoyant forces by the ocean."

P4, line 27: Replace "equivocal" with "controversial" and give reference(s) for this statement.

P5, lines 4-5: Rephrase to "Ice-marginal features form within the grounding zone, the transition from permanently grounded ice to ice that decoupled from its bed to become a floating ice shelf."

P5, line 5: Either mark listed features as examples (e.g. GZWs, marginal moraines, …) or list all of them.

P5, lines 11-12: Not exclusively – large GZWs may also indicate higher sediment flux.

P5, lines 13-14: Also reference Dowdeswell and Fugelli (2012), GSA Bulletin in this context.

P5, line 14: Replace "stratification is" with "reflectors are".

P5, lines 20-21: They were also described with clearly asymmetric shapes (cf. Winkelmann et al., 2010, QSR; Klages et al., 2013, QSR).

P6, line 2: Reference Larter et al. (2012), QSR and Klages et al. (2015), Geomorphology in this context. They described and interpreted those features.

P6, line 5: Rephrase to "Their association with vertical tidal movement…".

P6, line 17: Rephrase and avoid "propelled".

P6, line 21: Be more specific here and write: "Shelf-edge gullies on high-latitude continental margins…" since gullies are found on most continental margins.

P7, line 8: Write "… any potentially pre-existing subglacial landforms."

P8, line 6: Replace "monopolize" with "dominate".

P8, lines 9-10: Not only – they mainly imply bulldozing of proglacial debris.

P8, lines 15-16: The GZWs are large but not visible in the bathymetric data? Do you mean wide but relatively thin so that they are hardly visible in the bathymetry? Specify here.

The same applies for lines 25-27.

P8, lines 29-30: Rather write something like: "Phases of different flow directions in the ERS can clearly be identified by the presence of multiple generations of overprinting linear features."

P9, line 2: Replace "ensure" with "proof".

P9, line 3: Rephrase to "… each flowset only slightly deviate by less than $10°$, thus are assumed to represent…".

P9, line 4: Replace "isochronously" with "simultaneously" or "contemporaneously".

P9, line 5: Delete "after" and replace with "cf.".

P9, line 5-7: Modify sentence to "Assuming that all flowsets were shaped during and subsequent to the LGM, a relative chronology of their formation can be assessed based on their landward succession and cross-cutting relationships with other flowsets".

P9, lines 7-8: Modify sentence to "In order to characterize large-scale regional flow patterns, discrete flowsets within the $10°$ deviation are assumed to reflect a similar ice-flow configuration."

P9, lines 9-10: Modify sentence to "Our new compilation of RSE bathymetric data reveals that major flow patterns in the ERS generally deviate from the trough-parallel drainage that was described previously (…)."

P9, lines 10-12: Modify to "Flow in Glomar Challenger Basin may have been only partially parallel to the trough axis. We propose that a distinct cluster of linear features indicates flow also across an inter-ice stream ridge towards Whales Deep."

You sometimes write "Whales Deep" or "Whales Deep trough" – be consistent.

P9, line 12: Replace "curved" with curvilinear".

P9, lines 13-15: Modify sentence to "For those, rose diagrams were used to exclude the possibility the curvature indicates two discrete flow events with very similar orientations."P9, line 16: Replace "mirror" with "resemble".

P9, line 17: Replace "display generally" with "record".

P9, line 18: Write "…to the trough axis, pointing towards…".

P9, line 20: Replace "as" with "to indicate".

P9, lines 24-25: Rephrase to "We interpret the LGM grounding line in outer Drygalski Trough to have been situated just north of Coulman Island, marked by the outermost GZW (cf. Shipp et al., 1999)."

P9, line 26: Replace "field" with "cluster" and give reference to figure here.

P9, lines 26-27: Specify here. Recorded MSGLs rather give local evidence and provenance gives regional information.

P9, line 29: Rephrase to "In JOIDES Trough, maximum ice extent is suggested to be recorded by the large-scale GZW (J1) on the mid-outer shelf (Fig. 3)".

P9, lines 29-30: Rephrase to "We base this hypothesis primarily on the presence of an up to 8m-thick glaciomarine drape in the outer trough (Fig. 3a).

Do you have evidence for this statement (cores)? If not, phrase more carefully here.

P9, line 30 – P10, line 3: Rephrase to "The observation of LGM-age carbonates on surrounding banks (…) and the presence of LGM-age tephra layers in glaciomarine sediments on the outer shelf (…) further support this assumption."

P10, line 3: Are you really sure? Maybe these linear features are MSGLs as well but just represent an older ice advance prior to MIS2.

P10, lines 4-5: Rephrase to "The LGM limit in Pennell Trough coincides with the large-scale GZW (…), located ~120 lm landward of the shelf break (Howat and Domack, 2003).

P10, lines 7-9: Rephrase to "Large-scale GZWs at the shelf break, linear features that extend across the outer shelf, and extensive shelf-edge gullies (you could cite Gales et al., 2013, Geomorphology here) indicate that grounded ice likely reached the shelf break (…)."

P10, lines 9-10: Rephrase to "Thin glaciomarine sediments occur on the outer shelf and may hint at a relatively short period of ice-free conditions." Give reference for this statement and consider the possibility that post-LGM strata may be thin but was strongly reworked by scouring or winnowing, especially on outer shelves. This weakens your argument. Same with thick glaciomarine drapes – in some locations (e.g. Palmer Deep) they are extremely thick and people assumed that this drapes started to deposit well before the LGM but then they realized it was just of Holocene age. If you don't have radiocarbon ages, phrase more carefully.

P10, lines 11-12: You could easily combine Figs. 3 and 5. Try to implement Fig. 3 with Fig. 5 and rephrase sentence to "Figure 3 shows ... directions based on the appearance of linear features and GZWs."

P10, lines 15-16: Rephrase to "Bathymetric records from the WRS only revealed sparse and isolated patches of linear features that hamper meaningful interpretations of former subglacial flow behaviour and direction."

P10, line 22: Replace "ice-marginal features" with "moraines of GZWs".

P10, line 23: Use "seafloor" rather than "seascape" and delete "grounded".

P10, line 28: Give reference for this statement.

P11, lines 1-2: Same here and cite for example Smith et al., 2009, Quaternary Research in this context.

P11, line 3: Replace "scattered" with "a few".

P11, line 5: Replace "to the east and the west" with "east- and westwards".

P11, line 6: Say "during general deglaciation".

P11, line 6: Rather say "retreat behaviour" instead of "retreat pattern" and "dominated" rather than "dictated".

P11, line 8: Not the GZWs and moraines back-stepped onto banks but the GL.

P11, lines 9-10: I don't understand that. One main process of marginal moraine formation is the rain-out of debris-rich material proximal to the GL. So there is no need for flowing ice really in order to deposit a moraine. It can be a slight pushing effect but that could also origin from very slow concentric flow away from the bank. If you don't have lineations then you cannot say that there was a flow across this bank. Also consider reading Klages et al. (2013), QSR in that context.

P11, line 12: Rather write "These findings are supported by modelling approaches that...".

P11, line 15: Rather write "Reconstructed steps in GL retreat..."

P11, line 16: Replace "time-steps" with "phases" or "episodes" and delete "representing an interpretation of relative timing". It's redundant.

P11, line 20: Delete "back-stepping" and write "indicating a slowly retreating GL". Delete "in their path".

P11, lines 21-22: Clarify what you mean here. Why does drainage of an ice sheet nourishes an ice sheet?

P11, line 24: Rather write "unaffected by topography".

P11, lines 26-28: Write "Linear features on the ERS seafloor are overprinted by large-scale GZWs, indicating episodes of GL retreat that was interrupted by phases of temporary stabilization".

P11, line 29: Delete "behaviour".

P12, line 4: Replace "entails" with "requires".

P12, lines 6-8: Write "Trough-parallel flow was then established and ice began to retreat landward from the outer continental shelf in all ERS basins, interrupted by phases of stationary deposition of GZWs… ."

P12, lines 8-10: Write "Different generations of MSGLs are preserved as the GL retreats, but we would not … of streaming would have occurred, leaving one-directional MSGLs." Delete sentence in lines 9-10.

P12, line 14: Include "the presence of" between "on" and "large".

P12, line 15: Replace "curve across" with "flow onto neighbouring".

P12, lines 18-24: Write "Grounded ice retreated into Whales Deep to a mid-shelf location where it halted long enough…, before it retreated towards the inner shelf to halt again indicated by GZW e3".

How can an ice sheet not be in contact with the bed during retreat? That is contradicting. It would be an ice shelf then. And also an ice plain just prior to floatation may still create MSGLs. But an absence of subglacial landforms such as MSGLs does definitely not imply the absence if grounded ice. Rephrase completely and give alternatives with sufficient references.

P12, lines 25-27: Flow and thus subglacial sediment deposition on inter-ice stream ridges has recently been described as being very low (Klages et al., 2013, QSR). Cold-based ice may characterize them that is strongly coupled to its bed. At least include this different interpretation in your discussion.

P13, line 1: Delete "each", replace "sequence" with "succession" and write "…of events that could be tested if a greater coverage of …Basin would be available. It could illuminate cross-cutting … flowsets that are crucial … ."

P13, lines 3-6: Write "Additional multibeam bathymetric surveys of inter-ice stream ridges would also provide a better understanding of the general flow pattern (cf. Klages et al., 2013, QSR). Furthermore, reliable radiocarbon ages constraining min. GL retreat on the Whales Deep inner shelf would provide the only possibility in order to give evidence for early retreat and the formation of a 'long-lived' grounding-line embayment."

P13, line 10: Replace "highly" with "often" and give a few of those references.

P13, line 16: Replace "north-to-south" with "southward".

P13, lines 22-23: Write "This study but also previous studies suggest an initial significant GL retreat within the northern Drygalski Trough are consistent … recession of the GL … ."

P13, line 24: Write "…GL retreat on the remaining Ross Sea shelf (Fig. 7)…."

P13, line 25: Replace "calls for the persistence" with "suggests a persisting". It's actually not really clear what you try to say here. Try to rephrase.

P13, line 32: Are the ages from those publications rather ambiguous? If yes, rather delete sentence. If you keep it, you have to say why it is 'complicated' and give references for that statement.

P14, line 1: Include "geophysical" after "marine".

P14, line 16: Replace "seascape" with "seafloor".

P14, line 20: You could cite Larter et al. (2009) and Graham et al. (2009) here.

P14, lines 28-29: Which features transition into MSGLs?

P14, line 31: Delete "ages and".

P15, line 1: Replace "lithification" with "consolidation".

P15, line 6: Rephrase and give reference.

P15, line 8: Delete "Complex".

P15, lines 12-15: This passage is unclear. Rephrase.

P15, lines 20-23: You need to mention also the external forcings for GL retreat here such as atmospheric, ocean, and ice characteristics. I have the feeling that this should also go into the introduction.

P15, line 26: Delete "recessional" and write "…geomorphic features that indicate episodic, …". However, rapid and episodic are contradictory. Either specify and write that rapid retreat was interrupted by episodes of stillstand or just write "episodic retreat" (cf. Dowdeswell et al., 2008, GEOLOGY).

P15, line 28: Replace "over" with "across".

P15, line 29: Replace "indicators" with "features".

P15, line 30: Replace "was" with "has likely been".

P16, line 1: Add "shelf" behind "Ross Sea".

P16, line 1: Use inverted commata for the two model names.

P16, line 2: Replace "in the Ross Sea" with "here".

P16, line 5: Write "Ross Sea shelf".

P16, line 7: Replace "can be" with "are likely".

P16, line 8: Replace "wasn't" with "was not".

Add to the conclusions that there is a STRONG need for sediments and radiocarbon ages in order to verify or reject a lot of your hypotheses.

Tables and figures

Table 1: Since this table is very long, it could also go into the supplementary material. If you decide leaving it, please also add a column "Reference/data access", if available.

Figure 1: Indicate location of Fig. 2a. Delete the left arrow pointing away from "WAIS". Use the northward orientation of this figure for all the following figures. Include scalebar. In the figure caption add "et al." to the Fretwell reference.

Figure 2: Use orientation from Fig. 1 for clarity. Use white font color also for "i" in Fig. 2a. Also give scalebar. In Fig. 2b give vertical and horizontal scalebars in metres. Give

colorscales and north arrows for all the panels. Alternatively use orientation of Fig. 1 also here and say in caption that they are all S-N oriented. Give a reference for your statement that the basin is composed of soft deformation till.

Figure 3: Give lateral scalebars and increase font size. Give scalebar for main panel.

Figure 4: Indicate location of this figure in Figure 3 and give scalebar.

Figure 5: Dismiss this figure and combine with Figure 3. Also change references in text accordingly.

Figure 6: Give scalebar and again, be consistent with figure orientations. Take Figure 1 as basis for all other figures. Is there radiocarbon evidence for the scenario of ice persisting on banks?

Figure 8: Give scalebars. Are the colorscales in 'b' and 'c' the same as in 'a'? Indicate location of profile more obvious.

Figure 9: Use smaller and white arrows. The different colors for the locations of lineations and GLs are enough.

Figure 10: Very nice figure.

Figure 11: Give scalebar.

---

## Short Comment (SC1) · 17 Mar 2016

This is a very interesting study. The comments below might help improving the manuscript.

P4, L12-16: The definition of drumlin is incorrect. It is incorrect to define drumlins as downstream-tapering forms – morphometric data indicates that the majority of drumlins is symmetric and that drumlins wider/steeper on their lee side are as common as drumlins with wider/steep stoss than lee (see Spagnolo et al., 2009, 2011, 2012). Drumlins are defined in the paper as being over 10m high; morphometric data shows that drumlins less than 10m high are very common.

P4, L12-16: Defining drumlins as "moulded sedimentary landforms" is reductionist and not self-explanatory. Drumlin genesis is not fully understood and there is a range of putative mechanisms, some erosional, some depositional, etc. For example, see Eyles et al., 2016 (Sed. Geology).

P4, L12-16: Following the previous comments, the rationale for grouping drumlins with crag-and-tails (asymmetric, downstream-tapering forms) is invalid. Only some drumlins are crag-and-tails or crag-and-tail-like. A genetic definition for crag-and-tails would be useful.

P4, L14-16: Sentence seems to imply that crag-and-tails are not "moulded sedimentary landforms" (that with internal structure data it would be possible to differentiate between drumlins and crag-and-tails). Reword considering previous comments.

Section 3 (Methodology): Main focus should be on the paleoglaciological reconstruction framework. Geomorphological features, their classification and glaciological significance would be presented under Methodology (instead of results), such as in a table. The same applies to other concepts that appear for the first time in the Results (e.g., flowset).

Section 4 (Results): Consider re-organizing; for example, sub-sections 4.4 and 4.5 focus on geographical areas, whereas 4.1 to 4.3 present landforms. Using more levels (e.g., grouping landforms under a specific header) could help. Some of the information presented under Results would better come under Introduction and Methodology. See previous comment.
* * *

---

## Author Comment (AC1) · 26 Apr 2016

**Response to interactive comments on:**
**"Past Ice-Sheet Behaviour: Retreat Scenarios and Changing Controls in the Ross Sea, Antarctica" by A. R. Halberstadt et al.**

*We would like to thank all reviewers for their constructive and insightful suggestions which have helped us to improve this manuscript. Reviewer's comments are shown in black font, and the author response is shown in red italics. Author's comments reference manuscript changes by line number, corresponding to the attached Track Changes document ('Paleodrainage_edits.pdf') appended below.*

**Response to reviewer N. R. Golledge:**

The submitted manuscript presents new and legacy marine geophysical data from the Ross Sea, Antarctica, and uses this to reconstruct the pattern of flow both at the LGM and during its retreat. The paper is very well-written and well illustrated, with clear conclusions that are robust with respect to the data presented. I have no problem in recommending that the paper be accepted with only a few minor edits.

My only gripe really is that there has been quite a lot of Ross Sea work published recently, and not all of it is acknowledged here. This is unfortunate, because the papers I'm thinking of lend considerable support to the interpretations presented by the present authors and so would nicely bolster their arguments. In particular it would be good to acknowledge McKay et al., 2016 (Geology), who came to very similar conclusions based on different data.

*A paragraph has been added to '2 Study Area' (P3, line 13 - P4, line 11), providing an overview of previous work done in this area. The work done by McKay et al (2016) is included in that discussion.*

Clearly I have a bias in this regard, but I think it would be good if the efforts of the modelling community were also acknowledged. It is often stated in introductions to 'empirical' studies that the new data will help 'constrain numerical models', indeed, the current authors do this in the very first sentence of the Abstract. But what is the point of modellers using these geological data, if the models they produce are then disregarded? Maybe sometimes the modelling can help with the geological interpretations, rather than the other way around.

There are of course many modelling papers out there, but I know for a fact that Golledge et al., 2012, 2013, and 2014 all mention that retreat most likely started first in the deeper parts of the outer Ross Sea, and that the pattern of retreat was a product of incoming fluxes from both EAIS and WAIS, and was highly dependent on the location of bedrock highs. To illustrate my point, I'm uploading a figure showing the modelled grounding-line positions from the simulations published in McKay et al 2016, overlain on Figure 7 of the submitted paper. Personally I see a considerable amount of agreement there, which is gratifying because it means the models are

getting something right!

*It's very exciting to see the convergence between recently modelled grounding lines (Golledge, McKay, DeConto & Pollard, etc.) and our new reconstructions from purely geological and geophysical data. A paragraph explicitly discussing the contribution of the modelling community and the agreement between recent models and our reconstruction has been added under section '5.4 Comparison with existing deglacial models' (P18, line 32 – P19, line 10).*

Anyway, I'm not insisting that the authors have to cite all these papers, but it would be nice to 'close the loop' in a sense and recognise that sometimes synergies between modellers and empiricists can allow a convergence of views that together really show how flawed the 'swinging gate' model is.

Other than this, I can't really find fault with the paper, so I commend the authors for doing a great job pulling the data together and hope to see this published soon.

[revised manuscript text omitted]

---

## Author Comment (AC2) · 26 Apr 2016

**Response to interactive comments on:**
**"Past Ice-Sheet Behaviour: Retreat Scenarios and Changing Controls in the Ross Sea, Antarctica" by A. R. Halberstadt et al.**

*We would like to thank all reviewers for their constructive and insightful suggestions which have helped us to improve this manuscript. Reviewer's comments are shown in black font, and the author response is shown in red italics. Author's comments reference manuscript changes by line number, corresponding to the attached Track Changes document ('Paleodrainage_edits.pdf') appended below.*

**Response to reviewer J. P. Klages:**

Halberstadt et al. combined new multibeam swath-bathymetric data with already existing bathymetric and seismic datasets to present an extensive and comprehensive view of ice sheet extent and retreat in the Ross Sea Embayment (RSE), Antarctica. On the basis of this new compilation the authors were able to reconstruct flow pathways and retreat dynamics during and subsequent to the Last Glacial Maximum across the entire RSE, which led to some new conclusions about the ice sheet history in that region. The paper is well written, easy to grasp, but sometimes slightly lengthy and repetitive. The figures support the text sufficiently, however some figures could be easily combined with others in order to provide more clarity, and to save space. Generally and after consideration of the edits suggested below, I would like to see this manuscript published as it provides a new and valuable combination of datasets that allow a detailed insight into the Ross Sea Embayment glacial history.

As the editor already pointed out, I would like to see a more detailed implementation of this work with previous work from the area, especially in the introduction. In particular, more recent papers such as Bart and Owolana (2012), QSR and McKay et al. (2016), GEOLOGY need to be considered in this regard. Extensive work has been performed in the RSE and the authors should point out more clearly what is known so far, how their new results fit into these previous results, and how their newly presented results complement and maybe change them.

*An extensive discussion of previous studies in the RSE has been added to '2 Study Area' (P3, line 13 - P4, line 11) and includes these references.*

I further encourage the authors to incorporate the results of previous modelling efforts in more detail (e.g. recent studies by Golledge et al.) in order to define synergies. This would reveal the progress already achieved, but would also highlight the need for necessary future work. Building onto that it should be emphasized how empirical future work in the area could focus in order to reduce existing data-model mismatches.

*A paragraph has been added addressing model-data convergence and mismatches under section '5.4. Comparison with existing deglacial models' (P18, line 32 – P19, line 10).*

The authors should further point out that sediments and reliable radiocarbon dates for min. GL retreat are urgently needed in order to verify their hypotheses. Since their interpretations are exclusively based on geophysical data, they should phrase much more carefully in many parts of the manuscript. The lack of age control should be the strongest motivation for future work in the area.

*A caveat reminding the reader that absolute radiocarbon ages are necessary for constructing an absolute retreat history is now mentioned in '2 Study Area' (P4, lines 11-12), '5.4 Comparison with existing deglacial models' (P19, lines 9-10), and reiterated in '6 Conclusions' (P21, lines 21-24).*

*Geomorphic studies and radiocarbon dates complement each other; ages are required to integrate these reconstructed grounding-line patterns with larger-scale modeling efforts and previous work on Ross Sea deglacial chronology, but geomorphic context is also critical for interpreting radiocarbon dates and sediment facies. We feel that interpretations of relative timing can be reliably extracted from this comprehensive geomorphic dataset, giving it stand-alone value, yet reliable radiocarbon ages will increase the meaningfulness of this data in a larger context. This interrelationship is alluded to on P16, lines 13-15 and lines 18-19.*

Lastly, the introduction should emphasize the significance of the RSE for Antarctic ice sheet stability in more detail, maybe also in regard to other large embayments such as the Weddell and Amundsen Sea Embayment. Therefore, the significant contributions by The RAISED Consortium (2014) should be incorporated.

*Antarctic-wide context for this study has been included in '1 Introduction' (P2, lines 2-4).*

And – but this is just my personal opinion – I would suggest to slightly change the title of the manuscript to "Past Ice-Sheet Behaviour in the Ross Sea Embayment, Antarctica: Retreat Scenarios and Changing Controls".

*It is our hope that this paper will appeal to a wide and diverse audience, beyond just the scientists mostly focused on the Ross Sea. We hope that the existing title will attract modelers and workers studying glacial stability, in addition to Ross Sea-specific scientists. Thus, the Ross Sea location descriptor is relegated to the subtitle.*

If the aforementioned issues and the minor edits and suggestions in the supplementary file will be met sufficiently, I fully support the publication of this manuscript in C2 "The Cryosphere".

Technical corrections, suggestions for improving the readability, and some concerns from my side are listed in a supplementary file by page and line number.

*Indicates that the change has been corrected: **

P1, line 8: Replace "on" with "for" (*for* numerical ice-sheet models)  *

P1, line 12: Delete "in contact with the bed"

*Replaced sentence with: "Recessional geomorphic features in the WRS indicate virtually continuous back-stepping of the ice-sheet grounding line." (Instead of 'indicate virtually continuous retreat of the ice sheet in contact with the bed). (P1, lines 12-13)*

P1, line 22: Change to "The Ross Sea Embayment (RSE) drains ~25% of the AIS into the Ross Sea and thus is the largest drainage basin in Antarctica, fed by multiple ice streams…". *

P2, line 11: Change to "Multibeam swath bathymetry provides a record of bed conditions beneath the former ice sheet, …".   *

P2, line 12: Change to "These landforms record flow behaviour and past thermal regimes of formerly grounded ice."

*Corrected, although this paper does not address the implications of landform formation under different thermal regimes and therefore that term was omitted.*

P2, lines 15-16: Change to "This unique and integrated dataset … much higher resolution, thereby revealing the palaeo-ice sheet bed with a much higher resolution compared to their modern counterparts."

*Sentence now states (P2, lines 22-24): "This unique, integrated dataset provides an opportunity to view the paleo ice-sheet bed at a much higher resolution than is possible beneath the modern ice shelf and ice sheet." The original sentence was upheld in order to use the phrase 'much higher resolution' only once in the sentence.*

P2, lines 17-19: Change to "…this dataset to define glacial geomorphic features that characterize past flow and retreat dynamics, thus reconstruct ice-sheet paleodrainage across the Ross Sea Embayment during and subsequent to the LGM."

*Corrected, with the word 'define' changed to 'identify' since this study maps and interprets pre-defined geomorphic features.*

P2, line 21-22: "Change to "…, which preferentially eroded along pre-existing tectonic lineaments (you may give a reference here)."

*Corrected; the Cooper et al., 1991 reference applies to this statement as well.*

P3, line 5: Write "Austral summer".   *

P3, line 16: Replace "cannibalized" with "eroded" or "obliterated".   *

P3, line 24: Replace "post-LGM" with "postglacial", since some features may be covered by sediments that started to deposit prior to the LGM.   *

P3, lines 24-25: Replace "(post-)LGM" with "glacial" since some of the subglacial features in the RSE do not necessarily record LGM ice cover.

*For clarity, these features are now described as having 'formed during the last glacial cycle' (P5, line 12).*

Results section:

I suggest renaming section to "Results and interpretation".

*Descriptions of glacial geomorphic features have been moved up into the Methodology section, as suggested by the interactive comment from Marco G. Jorge (below).*

Descriptions of features and references to similar, already described features elsewhere are

largely missing – at least for the new dataset. Which landforms did you detect, how would you describe them, do they resemble already published features, and how do you interpret them on that basis (Description – Reference – Interpretation – Significance).

- *MSGLs:*
  - o *Description (P5, line 17, line 25)*
  - o *Ref (Clark, 1993; Tulaczyk et al., 2001; Shaw et al., 2008; Ó Cofaigh et al., 2008; Fowler, 2010; King et al., 2009; Anderson, 1999; Livingstone et al., 2012; Stokes and Clark, 1999; Shipp et al., 1999; Ó Cofaigh et al., 2002; Dowdeswell et al. 2004; Spagnolo et al., 2014, Heroy and Anderson, 2005)*
  - o *Interpretation (P5, lines 18-20, lines 23-27)*
  - o *Significance (P5, lines 21-22)*
- *Drumlinoid:*
  - o *Description (P6, line 1, lines 12-14)*
  - o *Ref (Benn and Evans, 2010)*
  - o *Interpretation & Significance (P6, lines 3-5)*
    - ▪ *Interpretation & Significance is discussed further in the context of the Ross-Sea-specific drumlinoid features (P11, line 18-23)*
- *Subglacial channels:*
  - o *Description (P6, line 19-20)*
  - o *Ref (Lowe and Anderson, 2003; Anderson and Fretwell, 2008; Smith et al., 2009; Nitsche et al., 2013; Witus et al., 2014; Alonso et al., 1992; Wellner et al., 2006; Greenwood et al., 2012)*
  - o *Interpretation & Significance (P6, line 23)*
    - ▪ *Interpretation & Significance is discussed further in the context of the Ross-Sea-specific channels (P10, lines 25-28)*
- *GZWs:*
  - o *Description (P7, line 5-6, lines 13-17)*
  - o *Ref (Alley et al., 1986, 1989; Anderson, 1999; Anandakrishnan et al., 2007; Alley et al., 2007; Dowdeswell et al., 2008, Dowdeswell and Fugelli, 2012; Batchelor and Dowdeswell, 2015; Shipp et al., 1999; Jakobsson et al., 2012; Simkins et al., in press; Anderson, 1999; Heroy and Anderson, 2005; Mosola and Anderson, 2006)*
  - o *Interpretation (P7, lines 6-8,)*
  - o *Significance (P7, line 6, lines 11-14)*
- *Marginal moraines:*
  - o *Description (P7, lines 25-27)*
  - o *Ref (Dowdeswell and Fugelli, 2012; Winkelmann et al., 2010; Klages et al., 2013; Batchelor and Dowdeswell, 2015; Hoppe, 1959; Lindén and Möller, 2005; Todd et al., 2007; Dowdeswell et al., 2008; Shipp et al., 1999; Jakobsson et al., 2011; Simkins et al., in press)*
  - o *Interpretation (P7, line 28 – P8, line 2)*

P3, lines 28--29: Change to "Subglacial landforms form beneath permanently grounded ice that exerts the offset buoyant forces by the ocean."

*Do you mean 'offsets the buoyant forces exerted by the ocean'? The first part of the change was made and the second half of the change was interpreted as described above (P5, lines 15-16).*

P4, line 27: Replace "equivocal" with "controversial" and give reference(s) for this statement.

*This sentence was altered to: '[channels have been observed…] although their origin and link to subglacial meltwater is not evident.' (P6, lines 23-24)*

P5, lines 4--5: Rephrase to "Ice--marginal features form within the grounding zone, the transition from permanently grounded ice to ice that decoupled from its bed to become a floating ice shelf."

*This has been corrected, with the exception of 'grounding zone' which has been replaced with 'grounding line'. We feel that the majority of the GZWs and certainly all of the marginal moraines were formed at a distinct and identifiable ice margin, and we are attempting to step away from the grounding 'zone' terminology unless there is evidence for a diffuse 'zone' rather than a 'line' (P7, lines 1-2).*

P5, line 5: Either mark listed features as examples (e.g. GZWs, marginal moraines, …) or list all of them.

*The ice-marginal features discussed in this paper (GZWs, marginal moraines, and linear iceberg furrows) are all listed in this sentence (P7, line 3).*

P5, lines 11--12: Not exclusively – large GZWs may also indicate higher sediment flux.

*This sentence now reads: "Large GZWs can imply longer episodes of stability…" rather than "Large GZWs mark longer episodes of stability..." (P7, line 13)*

P5, lines 13--14: Also reference Dowdeswell and Fugelli (2012), GSA Bulletin in this context.   *

P5, line 14: Replace "stratification is" with "reflectors are".   *

P5, lines 20--21: They were also described with clearly asymmetric shapes (cf. Winkelmann et al., 2010, QSR; Klages et al., 2013, QSR).

*References added, and sentence altered to reflect the existence of both symmetric and asymmetric marginal moraines (P7, lines 25-28).*

P6, line 2: Reference Larter et al. (2012), QSR and Klages et al. (2015), Geomorphology in this context. They described and interpreted those features.

*Klages et al. (2015) was added as a reference. Larter et al. (2012) proposes a conceptual model for the ploughing of linear features, but did not observe corrugation ridges in their study area, the Filchner Trough. Thus, they are cited in P9, line 1 accordingly.*

P6, line 5: Rephrase to "Their association with vertical tidal movement…".   *

P6, line 17: Rephrase and avoid "propelled".  *

P6, line 21: Be more specific here and write: "Shelf--edge gullies on high--latitude continental margins…" since gullies are found on most continental margins.   *

P7, line 8: Write "… any potentially pre--existing subglacial landforms."

*Corrected, with the exception of the word 'subglacial' since iceberg furrows could also overprint ice-marginal features*

P8, line 6: Replace "monopolize" with "dominate".   *

P8, lines 9--10: Not only – they mainly imply bulldozing of proglacial debris.   *

*Many processes have been invoked to form shelf-edge gullies, and it seems that the processes that cause these features is still uncertain and likely highly variable. This sentence (now P9, lines 15-17) has been altered to reflect the multitude of processes that could cause gully formation.*

*From Gales et al., 2012: "Antarctic gully formation has been attributed to erosion by: (1) mass flows, such as sediment slides, slumps, debris flows, and turbidity currents, with triggering mechanisms including resuspension by shelf and contour currents, gas hydrate dissociation, tidal pumping beneath large icebergs and near ice shelf grounding lines, iceberg scouring, tectonic disturbances, and rapid accumulations of glaciogenic debris at the shelf edge during glacial maxima [Larter and Cunningham, 1993; Vanneste and Larter, 1995; Shipp et al., 1999; Michels et al., 2002; Dowdeswell et al., 2006, 2008]; (2) subglacial meltwater discharge from ice sheet grounding lines during glacial maxima or deglacia- tions, whether by constant release [Wellner et al., 2001; Dowdeswell et al., 2006, 2008, Noormets et al., 2009] or more episodic and large-scale release [Wellner et al., 2006] possibly by meltwater evacuation from subglacial lakes [Goodwin, 1988; Bell, 2008]; and (3) dense water overflow [Kuvaas and Kristoffersen, 1991; Dowdeswell et al., 2006, 2008; Noormets et al., 2009]."*

P8, lines 15--16: The GZWs are large but not visible in the bathymetric data? Do you mean

wide but relatively thin so that they are hardly visible in the bathymetry? *Yes.* Specify here. The same applies for lines 25-27.   *

P8, lines 29-30: Rather write something like: "Phases of different flow directions in the ERS can clearly be identified by the presence of multiple generations of overprinting linear features."   *

P9, line 2: Replace "ensure" with "proof".

  *The word 'confirm' was used to replace 'ensure'.*

P9, line 3: Rephrase to "… each flowset only slightly deviate by less than 10°, thus are assumed to represent…".

  *Sentence now reads: "The orientation of linear features within a single flowset deviate by generally less than 10°, and thus each flowset is assumed to represent a single flow configuration whose component lineations were formed contemporaneously (cf. Clark, 1999)." (P12, lines 18-20)*

P9, line 4: Replace "isochronously" with "simultaneously" or "contemporaneously".   *

P9, line 5: Delete "after" and replace with "cf.".   *

P9, line 5-7: Modify sentence to "Assuming that all flowsets were shaped during and subsequent to the LGM, a relative chronology of their formation can be assessed based on their landward succession and cross-cutting relationships with other flowsets".   *

P9, lines 7-8: Modify sentence to "In order to characterize large-scale regional flow patterns, discrete flowsets within the 10° deviation are assumed to reflect a similar ice-flow configuration."

  *All of the linear features in each flowset generally fall within the ~10° deviation. This sentence is meant to convey that multiple flowsets were grouped together for ease of analysis. Groups of flowsets could (and do) span a range greater than the 10° deviation. Thus, this sentence (P12, lines 23-25) was altered for clarity to read: "In order to characterize large-scale regional flow patterns, flowsets with discrete yet similar orientations were assumed to reflect a similar ice-flow configuration and grouped together for analysis."*

P9, lines 9-10: Modify sentence to "Our new compilation of RSE bathymetric data reveals that major flow patterns in the ERS generally deviate from the trough-parallel drainage that was described previously (…)."

  *Corrected with modifications. The word 'generally' was replaced with 'often', since the dominant flow pattern in the ERS remains trough-parallel. Sentence now reads: 'Our new compilation of multibeam data reveals that major flow patterns in the ERS often deviate…' (P12, line 26)*

P9, lines 10-12: Modify to "Flow in Glomar Challenger Basin may have been only partially parallel to the trough axis. We propose that a distinct cluster of linear features indicates flow also across an inter-ice stream ridge towards Whales Deep."

  *Flow in GCB was at one time trough-parallel, and at another time across-trough. However, describing flow as 'partially parallel to the trough axis' implies that these two configurations coexisted. They perhaps did, to some extent, but that's not what the sentence*

*was meant to convey. This sentence was rewritten for clarity (P12, lines 28-30): "Some flowsets in Glomar Challenger Basin exhibit evidence of trough-parallel flow (flowsets a-c, Fig. 4), but other flowsets indicate flow across an inter-ice-stream ridge towards Whales Deep (flowsets d-h, Fig. 4)."*

You sometimes write "Whales Deep" or "Whales Deep trough" – be consistent. ✱

P9, line 12: Replace "curved" with curvilinear". ✱

P9, lines 13‑15: Modify sentence to "For those, rose diagrams were used to exclude the possibility the curvature indicates two discrete flow events with very similar orientations." ✱

P9, line 16: Replace "mirror" with "resemble". ✱

P9, line 17: Replace "display generally" with "record". ✱

P9, line 18: Write "…to the trough axis, pointing towards…". ✱

P9, line 20: Replace "as" with "to indicate". ✱

P9, lines 24‑25: Rephrase to "We interpret the LGM grounding line in outer Drygalski Trough to have been situated just north of Coulman Island, marked by the outermost GZW (cf. Shipp et al., 1999)." ✱

P9, line 26: Replace "field" with "cluster" and give reference to figure here. ✱

P9, lines 26‑27: Specify here. Recorded MSGLs rather give local evidence and provenance gives regional information.

*The allusion to till provenance was unnecessary and was removed.*

P9, line 29: Rephrase to "In JOIDES Trough, maximum ice extent is suggested to be recorded by the large‑scale GZW (J1) on the mid‑outer shelf (Fig. 3)". ✱

P9, lines 29‑30: Rephrase to "We base this hypothesis primarily on the presence of an up to 8m‑thick glaciomarine drape in the outer trough (Fig. 3a).

Do you have evidence for this statement (cores)? If not, phrase more carefully here. ✱

P9, line 30 – P10, line 3: Rephrase to "The observation of LGM‑age carbonates on surrounding banks (…) and the presence of LGM‑age tephra layers in glaciomarine sediments on the outer shelf (…) further support this assumption." ✱

P10, line 3: Are you really sure? Maybe these linear features are MSGLs as well but just represent an older ice advance prior to MIS2.

*These features are linear but do not have the extreme parallel conformity displayed by the ERS linear iceberg furrows. Thus, they are now described as 'straight furrows.' This explanation has been added to the text (P13, lines 25-26).*

P10, lines 4‑5: Rephrase to "The LGM limit in Pennell Trough coincides with the large‑scale GZW (…), located ~120 lm landward of the shelf break (Howat and Domack, 2003). ✱

P10, lines 7‑9: Rephrase to "Large‑scale GZWs at the shelf break, linear features that extend across the outer shelf, and extensive shelf‑edge gullies (you could cite Gales et al., 2013, Geomorphology here) indicate that grounded ice likely reached the shelf break (…)." ✱

P10, lines 9‑10: Rephrase to "Thin glaciomarine sediments occur on the outer shelf and may hint at a relatively short period of ice‑free conditions." Give reference for this statement and

consider the possibility that post-LGM strata may be thin but was strongly reworked by scouring or winnowing, especially on outer shelves. This weakens your argument. Same with thick glaciomarine drapes – in some locations (e.g. Palmer Deep) they are extremely thick and people assumed that this drapes started to deposit well before the LGM but then they realized it was just of Holocene age. If you don't have radiocarbon ages, phrase more carefully.

*Good point. Rephrased (P14, lines 4-5).*

P10, lines 11-12: You could easily combine Figs. 3 and 5. Try to implement Fig. 3 with Fig. 5 and rephrase sentence to "Figure 3 shows … directions based on the appearance of linear features and GZWs."

*Figs. 3 and 5 were combined and the sentence was changed accordingly.*

P10, lines 15-16: Rephrase to "Bathymetric records from the WRS only revealed sparse and isolated patches of linear features that hamper meaningful interpretations of former subglacial flow behaviour and direction."

*This rephrased sentence implies that patchy data coverage caused WRS linear features to appear sparse and isolated. However, bathymetric data coverage over the WRS is actually quite extensive; the linear features are described here as sparse and isolated because they were only observed in small patches either isolated in a few areas and mostly on the backs of GZWs, despite comprehensive coverage. The original sentence was slightly altered for clarity: "…the WRS contains sparse and isolated patches of linear features, providing only glimpses of subglacial flow behaviour and direction despite extensive multibeam data coverage. Therefore, most paleo-drainage interpretations in the WRS are based on ice-marginal features." (P14, lines 11-13)*

P10, line 22: Replace "ice-marginal features" with "moraines of GZWs".    *

P10, line 23: Use "seafloor" rather than "seascape" and delete "grounded".     *

P10, line 28: Give reference for this statement.

*The observation of the subglacial meltwater channels, and the interpretation that these channels were active during deglaciation, are not published; they stem from data collected during the recent NBP1502A cruise and are the subject of ongoing research within this group. Thus, this sentence (P14, lines 26-27) has now been changed to "We observe [meltwater channels…]".*

P11, lines 1-2: Same here and cite for example Smith et al., 2009, Quaternary Research in this context.

*See above.*

P11, line 3: Replace "scattered" with "a few".

*'Scattered' was replaced with 'isolated clusters of'*

P11, line 5: Replace "to the east and the west" with "east- and westwards".    *

P11, line 6: Say "during general deglaciation".

*This sentence was altered for clarification, to emphasize the implications of observing ice-marginal features in the deepest part of Central Basin – that ice remained grounded in Central Basin during deglaciation of that area, instead of lifting off the bed. Thus, the sentence*

*now reads: "…ice-marginal features in the Central Basin indicate that ice remained in frequent contact with the bed during deglaciation of this area." (P15, lines 4-5)*

P11, line 6: Rather say "retreat behaviour" instead of "retreat pattern" and "dominated" rather than "dictated".

*Corrected, although 'dictated' was replaced instead with 'controlled' not 'dominated'*

P11, line 8: Not the GZWs and moraines back-stepped onto banks but the GL.    *

P11, lines 9-10: I don't understand that. One main process of marginal moraine formation is the rain-out of debris-rich material proximal to the GL. So there is no need for flowing ice really in order to deposit a moraine. It can be a slight pushing effect but that could also origin from very slow concentric flow away from the bank. If you don't have lineations then you cannot say that there was a flow across this bank. Also consider reading Klages et al. (2013), QSR in that context.

*Good point – that is true for a moraine. However, we still observe highly asymmetric features that we interpret as GZWs on banks, calling for some sediment mobilization. The sentence was altered to reflect this (P15, lines 9-11).*

P11, line 12: Rather write "These findings are supported by modelling approaches that…".    *

P11, line 15: Rather write "Reconstructed steps in GL retreat…"    *

P11, line 16: Replace "time-steps" with "phases" or "episodes" and delete "representing an interpretation of relative timing". It's redundant.    *

P11, line 20: Delete "back-stepping" and write "indicating a slowly retreating GL". Delete "in their path".

*I don't think that the geomorphic observations in this area directly support the interpretation of slow GL retreat. In this area (southern Drygalski), retreat was likely relatively quick, based on the small sizes and repeating nature of back-stepping moraines and GZWs.*

P11, lines 21-22: Clarify what you mean here. Why does drainage of an ice sheet nourishes an ice sheet?

*This sentence now reads: "Drainage from the EAIS flowed into the Ross Sea Embayment until the last stage of deglaciation" – although it was moved to later in the text (P17, lines 25-28).*

P11, line 24: Rather write "unaffected by topography".    *

P11, lines 26-28: Write "Linear features on the ERS seafloor are overprinted by large-scale GZWs, indicating episodes of GL retreat that was interrupted by phases of temporary stabilization".

*This alteration was incorporated, and a secondary component was added to highlight the preservation of linear features and lack of small-scale recessional features. The sentence now reads: "Linear features on the ERS seafloor are overprinted only by large-scale GZWs (Fig. 3). These large-scale GZWs likely record periods of grounding-line stabilization, punctuated by episodes of ice sheet decoupling and grounding line retreat that back-stepped tens to hundreds of kilometres in distance and preserved linear features." (P15, lines 27-30)*

P11, line 29: Delete "behaviour".    *

P12, line 4: Replace "entails" with "requires".  *

P12, lines 6-8: Write "Trough-parallel flow was then established and ice began to retreat landward from the outer continental shelf in all ERS basins, interrupted by phases of stationary deposition of GZWs…."  *

P12, lines 8-10: Write "Different generations of MSGLs are preserved as the GL retreats, but we would not … of streaming would have occurred, leaving one-directional MSGLs." Delete sentence in lines 9-10.

*Corrected, although 'leaving one-directional MSGLs' was instead replaced with 'remoulding the bedform field.' Not sure what a one-directional MSGL is.*

P12, line 14: Include "the presence of" between "on" and "large".  *

P12, line 15: Replace "curve across" with "flow onto neighbouring".  *

P12, lines 18-24: Write "Grounded ice retreated into Whales Deep to a mid-shelf location where it halted long enough…, before it retreated towards the inner shelf to halt again indicated by GZW e3".  *

How can an ice sheet not be in contact with the bed during retreat? That is contradicting. It would be an ice shelf then. And also an ice plain just prior to floatation may still create MSGLs. But an absence of subglacial landforms such as MSGLs does definitely not imply the absence of grounded ice. Rephrase completely and give alternatives with sufficient references.

*This sentence was removed.*

P12, lines 25-27: Flow and thus subglacial sediment deposition on inter-ice stream ridges has recently been described as being very low (Klages et al., 2013, QSR). Cold-based ice may characterize them that is strongly coupled to its bed. At least include this different interpretation in your discussion.

*This was added as a new sentence (P16, lines 21-22): 'During trough-parallel flow, ice grounded on inter-ice-stream ridges was likely sluggish and cold-based, and strongly coupled to the bed (Klages et al., 2013)' (i.e. before flow was routed over the inter-ice-stream ridge)*

P13, line 1: Delete "each", replace "sequence" with "succession" and write "…of events that could be tested if a greater coverage of …Basin would be available. It could illuminate cross-cutting … flowsets that are crucial … ."  *

P13, lines 3-6: Write "Additional multibeam bathymetric surveys of inter-ice stream ridges would also provide a better understanding of the general flow pattern (cf. Klages et al., 2013, QSR). Furthermore, reliable radiocarbon ages constraining min. GL retreat on the Whales Deep inner shelf would provide the only possibility in order to give evidence for early retreat and the formation of a 'long-lived' grounding-line embayment."

*Corrected, with slight changes (P17, lines 13-14): "marine radiocarbon dates constraining grounding-line retreat on the Whales Deep inner shelf might provide evidence for early retreat…"*

P13, line 10: Replace "highly" with "often" and give a few of those references.  *

P13, line 16: Replace "north-to-south" with "southward".  *

P13, lines 22-23: Write "This study but also previous studies suggest an initial significant GL

retreat within the northern Drygalski Trough are consistent … recession of the GL …"

*Sentence now reads: "This study and previous marine studies (Licht et al., 1996; Cunningham et al., 1999; Anderson et al., 2014) suggest early grounding-line retreat within the northern Drygalski Trough, consistent with the swinging gate model." (P18, lines 15-17)*

P13, line 24: Write "…GL retreat on the remaining Ross Sea shelf (Fig. 7)…."  *

P13, line 25: Replace "calls for the persistence" with "suggests a persisting". It's actually not really clear what you try to say here. Try to rephrase.  *

*Sentence now reads: "In particular, our marine-based reconstruction suggests persistent EAIS drainage into the WRS throughout deglaciation…"(P18, lines 19-20).*

P13, line 32: Are the ages from those publications rather ambiguous? If yes, rather delete sentence. If you keep it, you have to say why it is 'complicated' and give references for that statement.

*Sentence removed.*

P14, line 1: Include "geophysical" after "marine".

*Sentence altered (P18, lines 28-29), omitting 'geophysical marine [data]': "Neither the swinging gate nor the saloon door model incorporate observations from the continental shelf…"*

P14, line 16: Replace "seascape" with "seafloor".  *

P14, line 20: You could cite Larter et al. (2009) and Graham et al. (2009) here.  *

P14, lines 28-29: Which features transition into MSGLs?

*Drumlinoids transition into MSGLs. Clarified.*

P14, line 31: Delete "ages and".  *

P15, line 1: Replace "lithification" with "consolidation".  *

P15, line 6: Rephrase and give reference.

*Rephrased. Mosola and Anderson 2006 initially suggested this concept and are already cited here (P20, lines 12-15).*

P15, line 8: Delete "Complex".  *

P15, lines 12-15: This passage is unclear. Rephrase.

*Sentence now reads: "Grounded ice in Little America Basin flowed over its eastern bank and converged with an outlet glacier draining Marie Byrd Land (flowset n, Fig. 4). This flow pattern implies that at one point, Little America Basin was not able to drain all of the ice flowing into it and therefore some of that ice was forced eastward out of the trough." (P20, lines 20-23)*

P15, lines 20-23: You need to mention also the external forcings for GL retreat here such as atmospheric, ocean, and ice characteristics. I have the feeling that this should also go into the introduction.

*The entire paragraph (P20, line 31 – P21, line 7) was modified to read: "Physiography exerts a first-order control on regional ice stream flow and retreat dynamics, and seafloor geology plays an important subsidiary role in controlling ice behaviour. These controls influence regional retreat patterns; more localised ice behaviour is still under investigation.*

*Numerous other processes affect glacial dynamics, such as ice-shelf buttressing, sediment shear strength and ice-bed coupling, and subglacial meltwater (e.g. Boulton et al., 2001; Dupont and Alley, 2005; Stearns et al., 2008). External forcings such as tidal effects, circumpolar deep water incursion and under-melting of ice shelves, and atmospheric effects are also influential (e.g. Rignot, 1998; Zwally et al., 2002; Arneborg et al., 2012; Walker et al., 2013). Ross Sea retreat was asynchronous between troughs, suggesting differential responses to these processes. Ongoing work on characterizing Ross Sea glacial geomorphology highlights the effect of these forcings on local grounding-line stability."*

P15, line 26: Delete "recessional" and write "…geomorphic features that indicate episodic, …". However, rapid and episodic are contradictory. Either specify and write that rapid retreat was interrupted by episodes of stillstand or just write "episodic retreat" (cf. Dowdeswell et al., 2008, GEOLOGY).

*'Episodic' was removed.*

P15, line 28: Replace "over" with "across".  *

P15, line 29: Replace "indicators" with "features".   *

P15, line 30: Replace "was" with "has likely been".

*… 'was' replaced with 'is believed to have been'*

P16, line 1: Add "shelf" behind "Ross Sea".   *

P16, line 1: Use inverted commata for the two model names.  *

P16, line 2: Replace "in the Ross Sea" with "here".

*… 'in the Ross Sea' was removed*

P16, line 5: Write "Ross Sea shelf".  *

P16, line 7: Replace "can be" with "are likely".

*Replaced 'can be' with 'are'*

P16, line 8: Replace "wasn't" with "was not".  *

Add to the conclusions that there is a STRONG need for sediments and radiocarbon ages in order to verify or reject a lot of your hypotheses.

*This sentence was added to '6 Conclusions' (P21, lines 21-24): 'Additional analyses of Ross Sea continental shelf sedimentology and additional reliable radiocarbon ages marking grounding-line retreat are necessary to test and refine the deglacial patterns proposed here. A radiocarbon chronology will help integrate our grounding-line reconstruction with previous work done on Ross Sea deglacial history.'*

Tables and figures

Table 1: Since this table is very long, it could also go into the supplementary material. If you decide leaving it, please also add a column "Reference/data access", if available.

*The table will go into Supplementary Material.*

Figure 1: Indicate location of Fig. 2a. * Delete the left arrow pointing away from "WAIS". * Use the northward orientation of this figure for all the following figures. * Include scalebar. * In the figure caption add "et al." to the Fretwell reference. *

Figure 2: Use orientation from Fig. 1 for clarity. * Use white font color also for "i" in Fig. 2a. * Also give scalebar. * In Fig. 2b give vertical and horizontal scalebars in metres. * Give colorscales and north arrows for all the panels. * Alternatively use orientation of Fig. 1 also here and say in caption that they are all S~N oriented.

Give a reference for your statement that the basin is composed of soft deformation till.

*The figure caption now reads: "MSGLs (3-5 m in amplitude) on the inner shelf of Glomar Challenger Basin occur above a glacial erosional surface, imaged by the high-frequency seismic" since there are no cores correlated to this CHIRP image to support that interpretation.*

Figure 3: Give lateral scalebars and increase font size. Give scalebar for main panel. *

Figure 4: Indicate location of this figure in Figure 3 and give scalebar. *

*Location is indicated in Fig. 1 instead.*

Figure 5: Dismiss this figure and combine with Figure 3. Also change references in text accordingly. *

Figure 6: Give scalebar and again, be consistent with figure orientations. * Take Figure 1 as basis for all other figures. * Is there radiocarbon evidence for the scenario of ice persisting on banks?

*Ice persisting on banks is interpreted from geomorphological observations. To my knowledge, the only current radiocarbon evidence for ice persisting on banks is the Taviani et al., 1993 carbonates on the JOIDES and Mawson outer shelf bank-tops.*

Figure 8: Give scalebars. Are the colorscales in 'b' and 'c' the same as in 'a'?

*Yes. This was added to the figure caption.*

Indicate location of profile more obvious. *

Figure 9: Use smaller and white arrows. The different colors for the locations of lineations and GLs are enough. *

Figure 10: Very nice figure.

Figure 11: Give scalebar. *

[revised manuscript text omitted]

---

## Author Comment (AC3) · 26 Apr 2016

**Response to interactive comments on:**
**"Past Ice-Sheet Behaviour: Retreat Scenarios and Changing Controls in the Ross Sea, Antarctica" by A. R. Halberstadt et al.**

*We would like to thank all reviewers for their constructive and insightful suggestions which have helped us to improve this manuscript. Reviewer's comments are shown in black font, and the author response is shown in red italics. Author's comments reference manuscript changes by line number, corresponding to the attached Track Changes document ('Paleodrainage_edits.pdf') appended below.*

**Response to interactive comment Marco G. Jorge:**

This is a very interesting study. The comments below might help improving the manuscript.

P4, L12-16: The definition of drumlin is incorrect. It is incorrect to define drumlins as downstream-tapering forms – morphometric data indicates that the majority of drumlins is symmetric and that drumlins wider/steeper on their lee side are as common as drumlins with wider/steep stoss than lee (see Spagnolo et al., 2009, 2011, 2012). Drumlins are defined in the paper as being over 10m high; morphometric data shows that drumlins less than 10m high are very common.

*This work uses the identification of drumlinoid features to constrain ice flow direction. In order to achieve that goal, we must ensure that the observed features form subglacially, and that they indicate ice flow direction. Thus, the discussion of drumlinoids in this paper has been reduced to only what is necessary for this context-specific interpretation: first, a paragraph describing this large and variable class of features in Methodology ('3.1 Subglacial Features'), and second, a paragraph describing the features observed in the eastern Ross Sea and why they can be interpreted to indicate paleo ice-flow in the Discussion section ('4.2 Eastern Ross Sea').*

*Under Methodology: "Smaller scale streamlined landforms, with lengths hundreds of metres to a few kilometres, comprise a number of landform classes such as drumlins, crag and tails, and megaflutes. We group these landforms here as a single class of drumlinoids. While their internal composition can be difficult to determine in the marine environment, and their formation mechanisms remain uncertain, this family of landforms is widely and most simply taken to record the former ice flow direction (Benn and Evans, 2010). In Antarctica, drumlinoids are most often observed at the transition between crystalline bedrock and sedimentary deposits (Wellner et al., 2001, 2006; Graham et al., 2009)." (P6, lines 1-14).*

*Under Discussion: "The only drumlinoids observed in the Ross Sea occur on the inner shelf of Glomar Challenger Basin (Fig. 2c), covering ~300 km², and are associated with a near-surface occurrence of crystalline bedrock (Anderson, 1999; Shipp et al., 1999). Because these features are moulded predominantly from bedrock, they likely formed over multiple glacial cycles. They*

*do, however, exhibit highly uniform orientations (Fig. 2c) that are consistent with MSGL orientations seaward of the drumlinoids, indicating that the most recent phase of ice flow was likely responsible for the final drumlinoid shape." (P11, lines 18-23).*

C1 P4, L12-16: Defining drumlins as "moulded sedimentary landforms" is reductionist and not self-explanatory. Drumlin genesis is not fully understood and there is a range of putative mechanisms, some erosional, some depositional, etc. For example, see Eyles et al., 2016 (Sed. Geology).

*See above.*

P4, L12-16: Following the previous comments, the rationale for grouping drumlins with crag-and-tails (asymmetric, downstream-tapering forms) is invalid. Only some drumlins are crag-and-tails or crag-and-tail-like. A genetic definition for crag-and-tails would be useful.

*See above.*

P4, L14-16: Sentence seems to imply that crag-and-tails are not "moulded sedimentary landforms" (that with internal structure data it would be possible to differentiate between drumlins and crag-and-tails). Reword considering previous comments.

*See above.*

Section 3 (Methodology): Main focus should be on the paleoglaciological reconstruction framework. Geomorphological features, their classification and glaciological significance would be presented under Methodology (instead of results), such as in a table. The same applies to other concepts that appear for the first time in the Results (e.g., flowset).

*Geomorphic features have been moved from Results into the Methodology section. The subheading Flowsets, however, remain in Results because we consider that analysis to our interpretation of data, rather than a description of data.*

Section 4 (Results): Consider re-organizing; for example, sub-sections 4.4 and 4.5 focus on geographical areas, whereas 4.1 to 4.3 present landforms. Using more levels (e.g., grouping landforms under a specific header) could help. Some of the information presented under Results would better come under Introduction and Methodology. See previous comment.

*Changes suggested in the previous comment have been enacted, so the Results section contains only two subheadings: '4.1 Western Ross Sea', and '4.2 Eastern Ross Sea'. 'Flowsets' has been moved to under Eastern Ross Sea with a subheading 4.2.1.*

[revised manuscript text omitted]

---

## Author Comment (AC4) · 26 Apr 2016

Thank you for your comments. A compilation of all reviewer suggestions and author responses is attached here as a supplementary file ('ReviewerResponses_All.pdf'), which references the page and line numbers from the edited manuscript with Track Changes recorded (attached here as a separate supplementary file, 'Paleodrainage_edits.pdf').

A version of the edited manuscript without Track Changes formatting is also included here as a supplementary file, in order to view a 'clean' version of the edited draft (Paleodrainage_edits_withoutTrackChanges.pdf').

[Figure]

These three files are included in the 'Paleodrainage_ARH' zipped attachment.

Please also note the supplement to this comment:
http://www.the-cryosphere-discuss.net/tc-2016-33/tc-2016-33-AC4-supplement.zip

─────────────────────────────

---

## Editor Comment (EC1) · C. R. Stokes (Editor) · 29 Apr 2016

I would like to thank all those involved in the interactive discussion of this paper and the authors of the manuscript for carefully considering their comments and, in most cases, revising the manuscript. I'm delighted to recommend acceptance.